# Otolith chemoscape analysis in whiting links fishing grounds to nursery areas

Neil M. Burns [1✉], Charlotte R. Hopkins [2], David M. Bailey [1] & Peter J. Wright [3]

Understanding life stage connectivity is essential to define appropriate spatial scales for fisheries management and develop effective strategies to reduce undersized bycatch. Despite many studies of population structure and connectivity in marine fish, most management units do not reflect biological populations and protection is rarely given to juvenile sources of the fished stock. Direct, quantitative estimates that link specific fishing grounds to the nursery areas, which produced the caught fish are essential to meet these objectives. Here we develop a continuous-surface otolith microchemistry approach to geolocate whiting (*Merlangius merlangus*) and infer life stage connectivity across the west coast of the UK. We show substantial connectivity across existing stock boundaries and identify the importance of the Firth of Clyde nursery area. This approach offers fisheries managers the ability to account for the benefits of improved fishing yields derived from spatial protection while minimising revenue loss.

[1] Institute of Biodiversity, Animal Health and Comparative Medicine, University of Glasgow, Glasgow G12 8QQ, UK. [2] Department of Biological and Marine Sciences, University of Hull, Hull HU6 7RX, UK. [3] Marine Scotland Science, Ecology and Conservation Group, Aberdeen AB11 9DB, UK. ✉email: neil.burns@sruc.ac.uk

Life stage connectivity is a critical process in the ecology of any species. Species resilience and survival often depend on dispersal to areas with suitable conditions and resources required at particular times during the lifecycle. Therefore, understanding life stage connectivity is important for spatially explicit management of wildlife populations[1]. Many aquatic organisms undertake ontogenetic migrations during their lives. Among fish species, these movements can be solely within freshwater systems, exclusively in marine environments and between freshwater and marine systems[2]. These ontogenetic movements allow access to a variety of habitats and resources to satisfy needs for: shelter from predators at early life stages[3], different food requirements at different life stages[4] and access to particular habitat types for reproduction[5]. Defining the extent of these movements and quantifying life stage connectivity in fish species is essential to improve ecological understanding and design appropriate management and conservation measures. Studies of life stage connectivity have been used to examine whether fish stock boundaries reflect the scale of population processes and the spatial links between larvae, juveniles and adults[6]. Identification of source areas of juveniles to the adult population can be important in defining areas which may be vulnerable to depletion[7,8]. Protecting these source nursery areas is often part of fisheries management and implemented through fisheries closures or gear measures to reduce bycatch of under-sized fish[9].

Modelled and observed data have demonstrated that protecting areas of juvenile aggregation can have a substantial positive benefit in conserving fish stocks by reducing mortality among juveniles[10,11]. However, identifying areas to protect based only on the density of aggregations of juveniles does not account for the contribution these areas make to adult aggregations[12]. Dense aggregations of juvenile fish in nursery areas may be present because of high larval input, but these areas may have high natural mortality, low growth rates or movement away from the area may be difficult[13,14]. The value of nurseries, therefore, depends on the relative contribution of juveniles to the spawning adult population[12,15]. Quantifying the degree of connectivity between nursery areas and catch locations is needed to implement management strategies, which reduce mortality in juvenile fish and maintain viable adult populations. It could also avoid the unnecessary economic losses, which would occur from restricting activities in less important nursery areas[16].

While potentially benefiting fish stocks, closed areas can cause economic losses to fishers whose activities are restricted. Additionally, the direct beneficiaries of closed areas are often unknown when these benefits, in the form of increased fishing yield, are experienced in other regions[17]. The benefits of protected nursery areas may be experienced in a different location if adults reach catchable size following ontogenetic shifts in distribution. Therefore, it is only possible to identify the efficacy of nursery area protection, and allocate costs and benefits appropriately, if beneficiary catching locations can be matched to the nursery areas from which recruits originate. If areas where adult fish are caught can be linked to nursery areas, an assessment of which nurseries are most important to the health of the stock could also be made and inform which nurseries are most likely to need management action.

Connectivity among fish life stages can be explored using otolith element signatures because regional variation in water chemistry and temperature can affect the incorporation of certain elements into the otolith[18–20]. Physiological influences have also been shown to affect concentrations of various elements in endolymph and otoliths[21–24], potentially complicating their use to recover geographic or environmental information. However, as long as all sources have been adequately sampled and samples are cohort matched to avoid potentially confounding effects of interannual variability, the combined effects of physiology and environment may augment among-site differences in otolith microchemistry and thus, increase the power of this tool to reconstruct connectivity patterns[23]. Specifically, elemental concentrations in equivalent parts of the otoliths of juvenile and adult fish from the same year-class provide a natural tag and allow life stage movements to be inferred[16,25–27]. While not common, there are some examples where the use of information from otolith studies has influenced management boundaries and measures to reduce recruitment overfishing. For example, in 2014 the North Sea and West of Scotland haddock (*Melanogrammus aeglefinus*) stocks were combined following evidence for juvenile exchange inferred from otolith microchemistry analysis[12]. Haddock and whiting (*Merlangius merlangus*) stock assessments in the Irish Sea (ICES Division VIIa) have also been adjusted to exclude areas, which harbour fish more closely connected to areas further south[28].

Tag-recapture studies indicate that mature whiting have a small migration range with limited exchange even across stock areas, but otolith microchemistry suggests that juveniles may disperse more widely including exchange between the North Sea and West of Scotland[29]. Distribution modelling has suggested an ontogenetic shift from inshore nursery areas to offshore spawning grounds[30] to the west of the UK. However, spawning also occurs in inshore areas and the contribution of inshore nursery areas to offshore spawning grounds is unknown. There are also spatial differences in the distribution of fishing fleets. Fishers targeting Norway lobster (*Nephrops norvegicus*) are concentrated in inshore waters while the fleet targeting mixed demersal fisheries has operated further offshore since the 1990s following declines in gadoid abundance[31]. Juvenile gadoids occupying inshore waters, and whiting in particular[30], are subject to extensive bycatch from the *Nephrops* fishery[28]. Developing equitable and effective management strategies for multiple species under such circumstances is particularly challenging. Protection of fish in one area, at a critical life stage, will inevitably disadvantage one part of the industry while benefiting another. Importantly, not all nursery areas are equally productive, and some areas may consistently contribute more individuals to the population than others. To ensure both equitable and effective protection measures, it is essential to be able to account for population scale movements of a target species and the accrual of the potential benefits of these movements.

Here, we quantified connectivity between juvenile post-settlement age-0 and mature age-1 whiting to the west of the UK by developing an otolith microchemistry continuous assignment approach. Unlike previous methods that rely on non-spatially explicit classification methods like discriminant function analysis, which assigns individuals to source groups, this method applies a spatially explicit, continuous assignment approach using otolith element concentrations. Traditionally, studies of population connectivity using otolith microchemistry have used classification methods to assign individuals to groups with similar elemental signatures. Classification methods suffer as the number of classes increases because the proportion of correct assignments naturally decreases with increasing class numbers[32]. One solution is to aggregate sampling sites[33], and by doing so generate predefined geographic regions. While this maximises classification accuracy, localised variation in element signatures and differences between the scale of variability among elements can confound group differentiation[34]. A continuous assignment approach does not require delineating boundaries between groups and better represents real-world spatial autocorrelation of element concentrations between sampling locations. The use of a continuous-surface assignment approach recognises that element

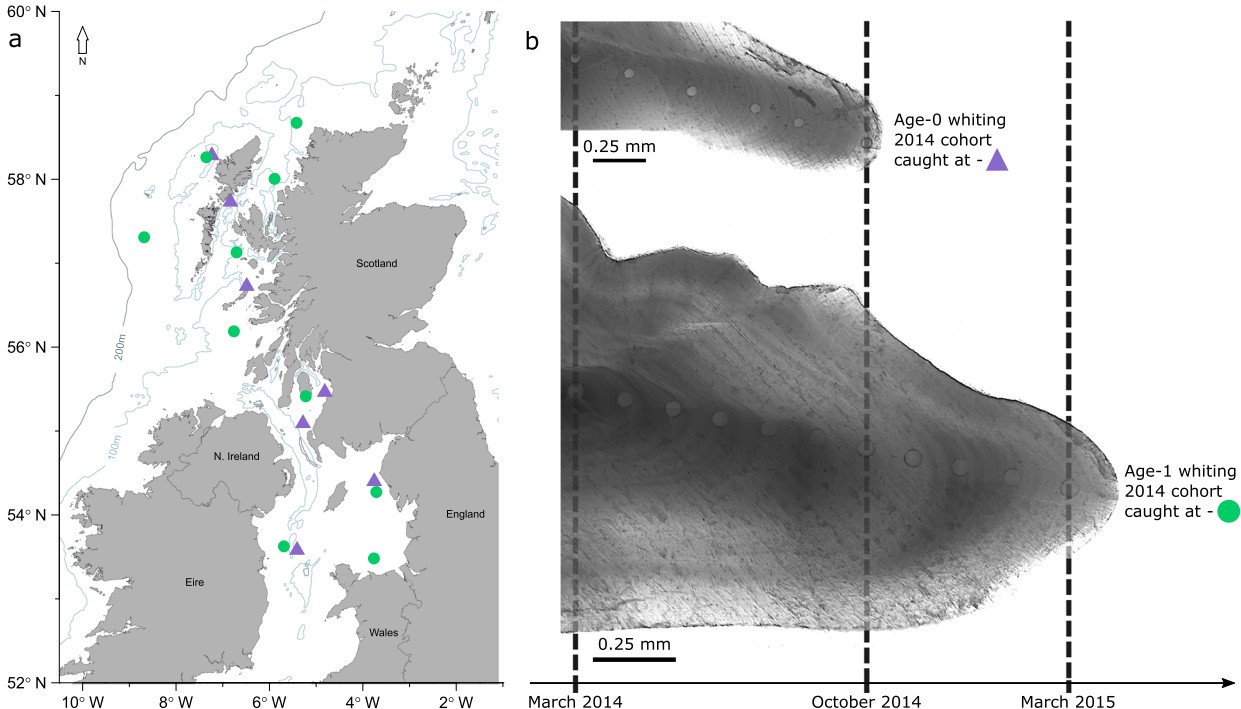

**Fig. 1 Sampling locations and temporal relationships between ablated otolith samples. a** Scientific trawl sample locations are shown for age-0 whiting caught in October 2014 (purple triangles) and age-1 whiting caught in March 2015 (green circles). **b** Otolith sections from October and March samples showing ablation pits. Dashed lines indicate corresponding time points across the otoliths. Dashed lines corresponding to otolith edges show sample trawl dates. The dashed line corresponding to October 2014 shows the pit location in age-0 fish otoliths used to construct the chemoscapes and train the geolocation model and, in age-1 fish to assign the age-1 fish to likely origin locations. x-axis arrow represents time.

| Table 1 Whiting otolith sampling details. | | | | |
| --- | --- | --- | --- | --- |
| **Sampling time** | **Number of otoliths sampled** | **Cohort, age** | **Sex ratio (M:F)** | **Contribution to analysis** |
| October 2014 | 140 (134) | 2014, Age-0 | 77:63 | Constructing and CV of continuous modelled element surface Oct. 2014 |
| March 2015 | 159 (155) | 2014, Age-1 | 139:20 | Continuous-surface assignment of age-0 origin at Oct. 2014 |

Number of otoliths sampled shows the total number sampled and, in brackets the number included in the final analysis. Samples were removed from the analysis that did not have ablation pits corresponding to the correct sample times. The skewed sex ratio (1:1 would have been expected) reported here in the age-1 whiting caught in March 2015 may result from this ontogenetic distribution change. Males often mature at an earlier age and so may also be more likely to move offshore to adult areas. Here, to allow valid within cohort comparisons of otolith chemistry, we targeted age-1 fish, which may result in the observed difference.

concentrations do not necessarily change abruptly across boundary lines. The ability to reconstruct the movements of fish from otolith microchemistry using a continuous assignment approach could improve investigations of fish movement ecology, connectivity and migration.

Although continuous assignment has been used successfully to geolocate several terrestrial and marine species from stable isotopes[35–37], these approaches have not before been paired with otolith microchemistry signatures to geolocate fish. Here, we develop a continuous-surface assignment approach applied to otolith microchemistry for the first time. However, unlike stable isotope ratios that have clearly defined mechanisms relating tissue signatures to ambient environmental conditions element incorporation into otoliths has been shown to be influenced by both environmental and physiological conditions[21,23], the resulting microchemical signature can nevertheless make an effective natural tag[34,38]. Otolith element concentrations were sampled from age-0 juvenile whiting and material at the corresponding time point from mature age-1 fish from the same cohort using Laser Ablation Inductively Coupled Plasma Mass Spectrometry (LA-ICP-MS). We modelled the spatial distribution of element

concentrations sampled from age-0 whiting otoliths as continuous-surface "chemoscapes." The most likely juvenile origins of age-1 fish were then predicted from these chemoscapes. From this analysis, population scale connectivity between life stages at ten adult catch locations across the UK west coast was inferred. These results reveal substantial connectivity across current stock boundaries and suggest that the Firth of Clyde is a critical nursery area supplying juveniles to fished aggregations in two stock areas.

## Results

**Trace element continuous-surface modelling.** Age-0 whiting collected in October 2014 by bottom trawl from seven sample locations across the west coast of the UK (Fig. 1a) were used to construct continuous-surface predictions of otolith trace element concentrations. The location of age-0 whiting trawl samples in the present study represent the areas identified as most persistently occupied by this age group documented in Burns et al.[30]. Elemental compositions from otolith material ablated near the edge of 140 age-0 whiting otoliths were used to construct the

**Table 2 Spatial GAM summaries.**

| Element | Explanatory variables | edf | F- statistic | p | Cross-validation (Spearman's rank correlation) |
|---------|----------------------|-----|--------------|---|-----------------------------------------------|
| Na | so(x, y) Gaussian | 1.92 | 0.79 | <0.01 | $\rho = 0.93$, $p < 0.01$ |
| Mg | so(x, y) Gaussian | 2.47 | 2.23 | <0.001 | $\rho = 0.86$, $p < 0.05$ |
| Ba | so(x, y) Gamma | 3.95 | 3.08 | <0.001 | $\rho = 0.82$, $p < 0.05$ |
| Sr | so(x, y) Gaussian | 5.55 | 4.23 | <0.001 | $\rho = 0.96$, $p < 0.01$ |
| Mn | so(x, y) Gamma | 4.53 | 2.94 | <0.001 | $\rho = 0.99$, $p < 0.001$ |
| Rb | so(x, y) Gamma | 4.46 | 2.22 | <0.001 | $\rho = 0.96$, $p < 0.01$ |

Explanatory variables retained during model selection and Spearman's roh ($\rho$) and p-values for the results of the spatial GAM cross-validation are displayed. "so" denotes soap-film smoother applied to spatial variables modelled using the indicated distribution.

chemoscapes (Table 1). Material from the same position close to the ventral edge of the age-0 whiting otoliths, accreted recently in the life of the fish provided a chemical signature at a known geographic location (i.e., the sample trawl sites from October 2014; Fig. 1). The use of laser ablation in ICP-MS techniques allows high temporal resolution sampling of otolith material. Even at the finest scales, laser ablation pits may account for several days in the life of fish. This study used a beam diameter of 55 μm, which equates to roughly 10 days in the life of this species. Tobin et al.[29] estimated the average displacement distance of whiting (at liberty >30 days) from historic tag-recapture data to be between 16 and 78 km depending on the time of year. Each "point" sample, calculated as the midpoint of the trawl, therefore, actually represents an area of uncertainty (presumably < 78 km) around the trawl location.

A suite of 17 elements ($^{7}$Li, $^{23}$Na, $^{24}$Mg, $^{27}$Al, $^{31}$P, $^{45}$Sc, $^{47}$Ti, $^{52}$Cr, $^{55}$Mn, $^{60}$Ni, $^{65}$Cu, $^{66}$Zn, $^{85}$Rb, $^{88}$Sr, $^{89}$Y, $^{138}$Ba, $^{208}$Pb) was selected to maximise the chance of observing spatial variability and sampled using LA-ICP-MS. Sr and Ba have been shown to vary with temperature and salinity[39,40]. Al, P, Cu and Pb may be present at higher concentrations where anthropogenic inputs raise their levels in the environment, and this may be reflected in the trace element concentrations found in otoliths from fish sampled in areas like the east Irish Sea and Firth of Clyde. Other elements were selected based on evidence from previous studies, which detected differences in trace elements in gadoid otoliths sampled in similar areas. For example, Mg, Mn, Rb and Ba[12,27,29,41,42] and Zn, Ni, Cr, Ti and Sc[43]. A further three elements, Li, Na and Y were included as they had been shown to improve classification success in other fish species[44].

The spatial variability of each element was tested individually by fitting a Generalised Additive Model (GAM) with a 2-dimentional soap-film smoother using longitude and latitude as explanatory variables. Of the initial17 elements considered six ($^{23}$Na, $^{24}$Mg, $^{55}$Mn, $^{85}$Rb, $^{88}$Sr, $^{138}$Ba) showed spatial variation (Table 2). The geographic variability of water chemistry for several of these six elements has been demonstrated previously, with more coastal waters showing higher variability than open ocean[45]. For example, water concentrations of Ba tend to increase with depth in marine systems[46], and particulates from inshore areas have been found to be enriched with Mn[47,48]. Beyond the immediate ambient elemental concentrations, the concentrations of elements like Ba and Sr in otoliths are influenced by factors like temperature and salinity[39,40]. Variations in Mn concentrations have been used successfully in several studies in UK waters[27,29,41,49]. Temporally persistent regional otolith signatures of Ba and Mn have also been documented occurring in UK waters[41]. While Mn and Mg have been recognised as a biomineralization cofactors involved in otolith formation[50,51], the concentration of Mn in otoliths has also been explained by extrinsic factors like hypoxia[52]. Although there are yet no

definitive explanations for the mechanisms governing observed differences in otolith concentrations of many elements, the spatial variation reported in the current study has allowed the construction of cross-validated chemoscapes from sampled otolith material. Table 2 shows the spatial GAM summaries and results of model cross-validation from tests on 1000 iterations of a 4:1 data split training set stratified across catch locations.

**Maximum likelihood geolocation model selection**. The six elements showing spatial variation were then tested as suitable candidates for inclusion in the maximum likelihood (ML) geolocation model. To optimise the geolocation model, we applied the methods of Vander Zanden et al.[36] and Trueman et al.[37] to define the accuracy and precision of each model for assigning catch locations to individual age-0 whiting. Described in detail in the methods, model selection was conducted with Monte Carlo cross-validations using the accuracy and precision of the model geolocation predictions to select the optimal predictive set of elements from the six spatially variable elements. Each round of model selection used the predicted catch locations of all age-0 whiting to define the model accuracy and precision by comparison to the known, true catch locations. The optimal predictive model containing Sr and Mn, was identified by a round of backward stepwise selection from a model containing all six spatially variable elements followed by a round of forward stepwise selection. Here, we defined the optimal predictive model as one, which maximised accuracy versus precision.

The modelled chemoscapes retained in the geolocation model (Sr and Mn) and their associated variances are displayed in Fig. 2. High concentrations of Sr were detected in otoliths from northern and western areas. The GAMs therefore predicted high concentrations of Sr around the Hebrides and also in the Western Irish Sea. Low Sr values were observed in samples and predicted by the GAM in the eastern Irish Sea. The areas of highest model uncertainty occurred in the western extent of the chemoscape, farthest from sample sites. Chemoscape Mn concentrations were highest in more southerly, eastern waters, in the Irish Sea and in the southern Firth of Clyde. The eastern Irish Sea showed particularly high values, but this is also associated with high uncertainty to the east of the Isle of Man.

**Predictive power of geolocation from chemoscapes**. The accuracy and precision of the optimal geolocation model that assigned the most likely catch locations to individual age-0 whiting from Sr and Mn otolith concentrations is displayed in Fig. 3. At all levels of precision (defined as 1- proportion of modelled area), the model showed better than random accuracy of assigning the correct origin location to an individual. The Area Under the Curve (AUC) was 0.95 and the model generated a mean linear error of 242 km between the predicted catch locations and the

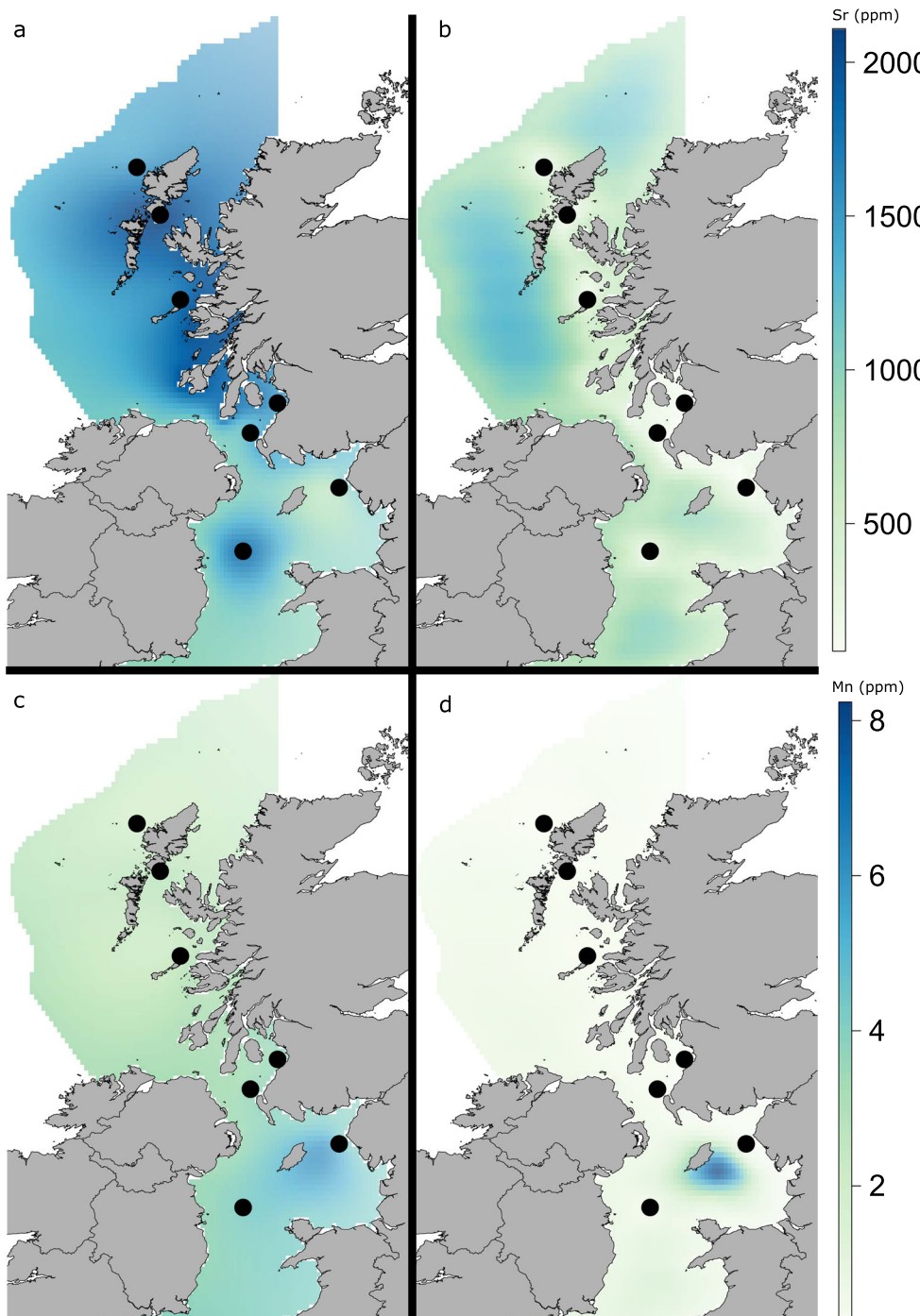

**Fig. 2 Chemoscapes predicted by GAM for March 2014 for Sr and Mn. a**, **c** Mean concentration (ppm) of Sr and Mn and, **b**, **d** standard error for Sr and Mn. Filled circles indicate sample trawl locations.

true catch locations. The baseline model produced high accuracy geolocation estimates for the true capture location of fish over the range of precisions. Accuracies of 90% were achieved at a precision of 8% of the total area. This model covered 404,715 km$^2$ of the UK west coast and, therefore, showed a spatial precision of 8094 km$^2$ equivalent to a circle with a radius 50.76 km.

**Continuous-surface assignment to geolocate age-1 fish.** The continuous-surface geolocation model, fitted and tested on the age-0 fish caught in October 2014 was then used to assign age-1 whiting, caught in March 2015 to their age-0 (October 2014) origin locations. We, therefore, predicted the age-0 juvenile origin

location of age-1 fish using material from the portion of age-1 otoliths that was accreted at the same time as the otolith edge material from the October 2014 juvenile phase. The combined results, used to make population scale inferences, compare the catch location of age-1 fish (in March 2015) to their predicted origin location at age-0 (in October 2014) for all individual fish caught at each sample trawl location are displayed in Fig. 4.

The likely juvenile origins, of mature age-1 whiting inhabiting the most northerly waters sampled off the Scottish west coast, North-North Minch (NNM) are widespread (Fig. 4a). Fish caught in this trawl sample were assigned origins from the Firth of Clyde (Cl), Northeast Irish Sea (NEIS), South Minch (SM) and Offshore

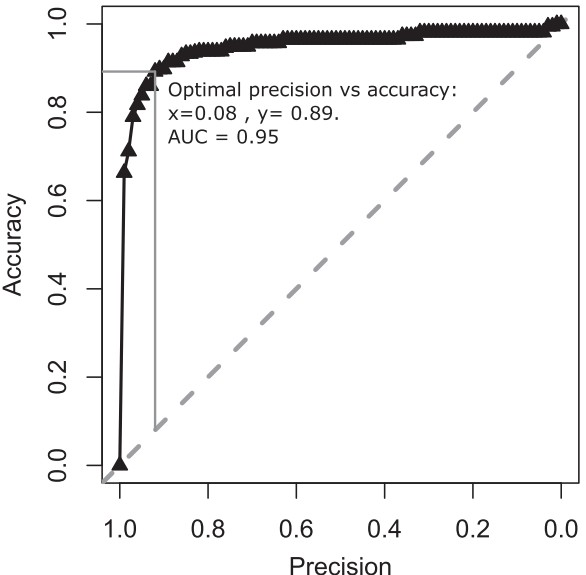

**Fig. 3 Precision and accuracy of otolith microchemistry assignment.**
Precision is defined as 1—the proportion of the total modelled area (i.e., smaller areas equate to higher precisions). Accuracy is the proportion of correct fish geolocations contained within the area specified by the precision. The diagonal dashed grey line represents the accuracy, which would be achieved by random allocation. The solid, grey vertical and horizontal lines show the optimal precision vs. accuracy values defined by (precision = 0.08, accuracy = 0.89).

(Off) to the west of the Hebrides. No age-1 fish caught at the NNM sample site were likely to have originated from nursery areas locally and were equally likely to be derived from any of the other juvenile areas highlighted. Three sample sites, namely Offshore (Off), West Offshore (WOff) and the South Minch (SM) showed similar connectivity to each other (Fig. 4b, d, e). Some age-1 fish caught at these three sites were shown to have local nursery origins. However, most juvenile fish in these trawl samples were likely to be derived from further south; from the Firth of Clyde (Cl) and Northeast Irish Sea (NEIS). Whiting caught in the North Minch (NM) were unlikely to be from local nursery areas and were much more likely to have are-0 origins in the Firth of Clyde (Cl) and Irish Sea (Fig. 4c). Age-1 whiting caught further south near the Inner Hebrides (IH) showed widespread recruitment from several areas across the UK west coast. The most likely origin areas for these age-1 Inner Hebrides (IH) caught fish include the Firth of Clyde (Cl), Northeast and West Irish Sea (NEIS & WIS) and areas further north on the Scottish West Coast like the North Minch and Offshore (NM & Off) (Fig. 4f). While Firth of Clyde (Cl) caught age-1 whiting were most likely to have been derived from age-0 fish from the Northeast Irish Sea (NEIS) they also show a high likelihood of originating from local nurseries (Fig. 4g). Eastern Irish Sea age-1 fish caught at both sample sites (NEIS & EIS) are more likely to be derived from local sources and show limited connectivity with other areas apart from the Firth of Clyde (Fig. 4h, j). Of the Irish sea samples the West Irish Sea (WIS) age-1 whiting showed the most widespread likely age-0 origins. While most West Irish Sea age-1 whiting were likely to derive from age-0 fish from the Firth of Clyde (Cl) and east Irish Sea (NEIS & EIS) or be derived from local nurseries they were also linked to Scottish west coast waters (Fig. 4i).

**Quantification of life stage connectivity.** The catch locations of the mature age-1 whiting were used to define ten Voronoi cells,

across the Scottish West Coast and Irish Sea. The probable origins of age-0, juvenile whiting recruiting to these age-1 catch locations are displayed in the matrix columns of Fig. 5. The matrix also describes the likely destinations for age-0 whiting supplying the ten, age-1 catch locations (Fig. 5a, rows).

The majority of age-1 mature whiting caught on the Scottish West Coast and in the Irish sea originated from nursery areas in Cl. Exceptions to this were observed in both eastern Irish Sea areas (NEIS & EIS). These Irish Sea areas were more closely linked to each other and were supplied predominantly from local nurseries. Looking down the matrix diagonal (Fig. 5), other than Cl minimal local recruitment was evident in any other of the Scottish west coast areas (NNM, Off, NM, WOff, SM & IH). Fish from the Cl, NEIS and EIS showed a high likelihood of being derived from local nurseries. These areas were also the likely sources for substantial numbers of juveniles supplying the wider Scottish West coast and Irish Sea. The right margin of Fig. 5 shows the strongest contribution of juvenile whiting to the adult populations on the Scottish West Coast and Irish Sea likely originate from nursery grounds in the Cl and NEIS.

The current stock boundary separates West of Scotland (ICES Division VIa) whiting with those in the Irish Sea (ICES Division VIIa). However, a clear link exists between these two Divisions. The overall contribution to the west of the UK is greater from the Irish Sea and Cl than all other areas. Several Scottish West Coast areas like the NNM, NM and SM are supplied by east Irish Sea juveniles (NEIS and EIS). Although the Cl shows a high proportion of fish derived from local nurseries it also harbours whiting derived from the NEIS. The results here highlight the importance of the Firth of Clyde as a whiting nursery and the strong link between the Irish Sea and the Scottish West Coast, two areas, which are currently assessed as separate stocks.

## Discussion

Our use of otolith microchemistry to generate a continuous likelihood surface allows the movements of individual fish to be inferred by geolocating fish at a previous time point in their life history. The methods developed here can be used to combine the results from several individuals to understand fish movements at the population level[36,37]. The plots of the likely origin of age-1 whiting from their age-0 juvenile life stage (Fig. 4) provide a validated and easily interpreted description of connectivity. This method allows the potential benefits of protecting areas important to juveniles to be quantified and shows to which areas these benefits accrue.

Using otolith microchemical signatures to geolocate fish and infer their movements requires element concentrations to vary spatially at appropriate scales. Three of the spatially variable elements described here (Ba, Sr and Mn) also feature in the work of Tobin et al.[29] and were sampled at similar times of the year, and from the same species. The otolith element concentrations described from their Scottish west coast samples show similar spatial trends to those observed here. For example, Mn concentrations were similarly high in areas further south and west. Similar spatial patterns for Mn otolith concentrations are observed in a related species, Atlantic cod (*Gadus morhua*)[27]. While some studies examining otolith trace elements have found temporal consistency to occur[53] in most however, temporal inconsistency is the norm and resampling to establish a baseline is required to study the same area in subsequent years[54,55].

Although it would be expected that trace element concentrations in sea water are a strong driver for the values reported from otoliths, metabolic control may also influence element deposition[20,22,23,56]. The interpretation of correlative links between the environmental variables trialled and element

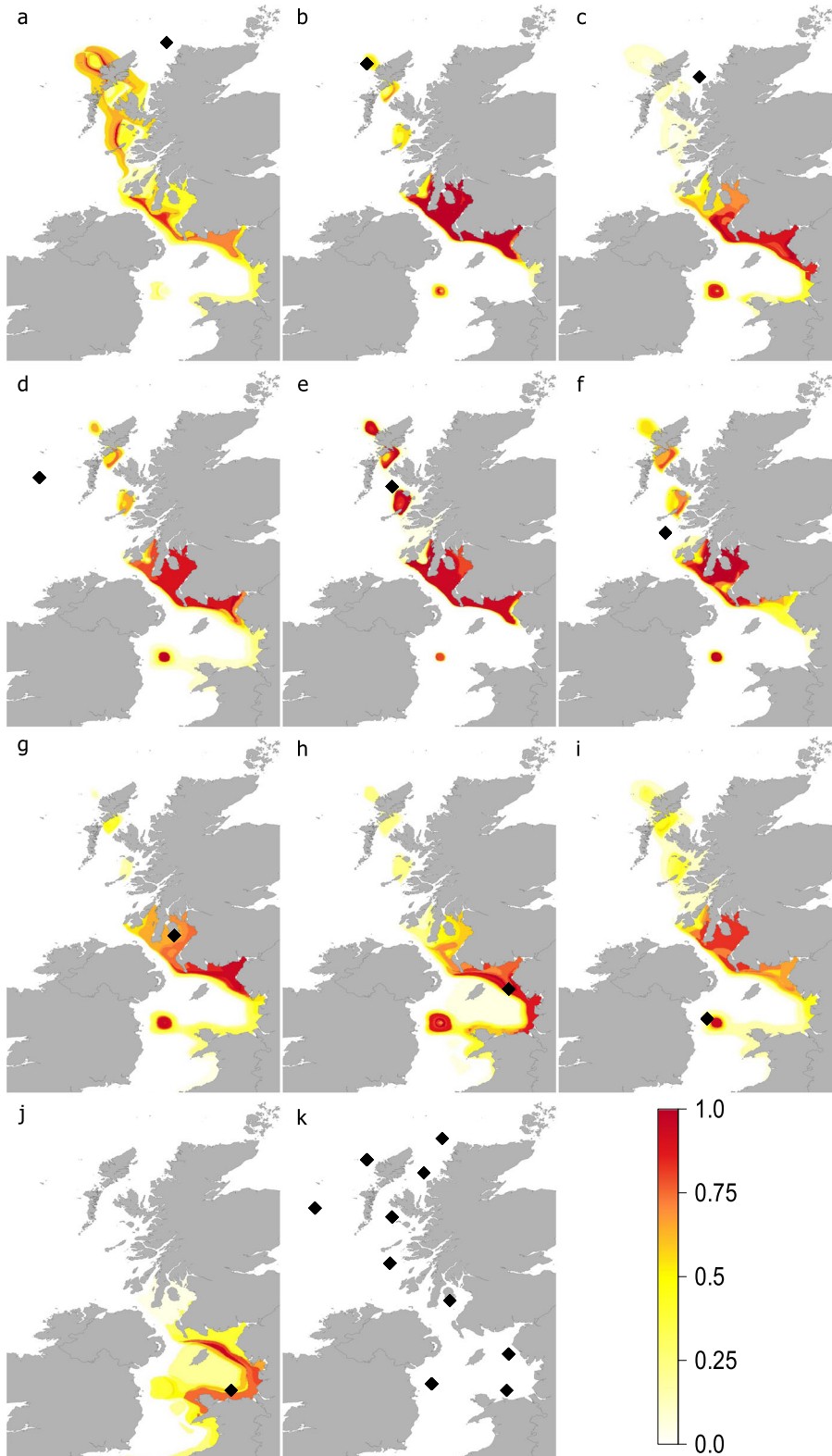

**Fig. 4 Age-0 origin of age-1 whiting.** Heat map shading shows the most likely age-0 origin (at October 2014) locations for mature age-1 fish caught in March 2015. Black diamonds indicate location where age-1 whiting were caught in trawl samples in March 2015, **a** North-North Minch (NNM), **b** Offshore (Off), **c** North Minch (NM), **d** West Offshore (WOff), **e** South Minch (SM), **f** Inner Hebrides (IH), **g** Firth of Clyde (CI), **h** Northeast Irish Sea (NEIS), **i** West Irish Sea (WIS) and **j** southern East Irish Sea (EIS). **k** displays the names for all trawl sample locations.

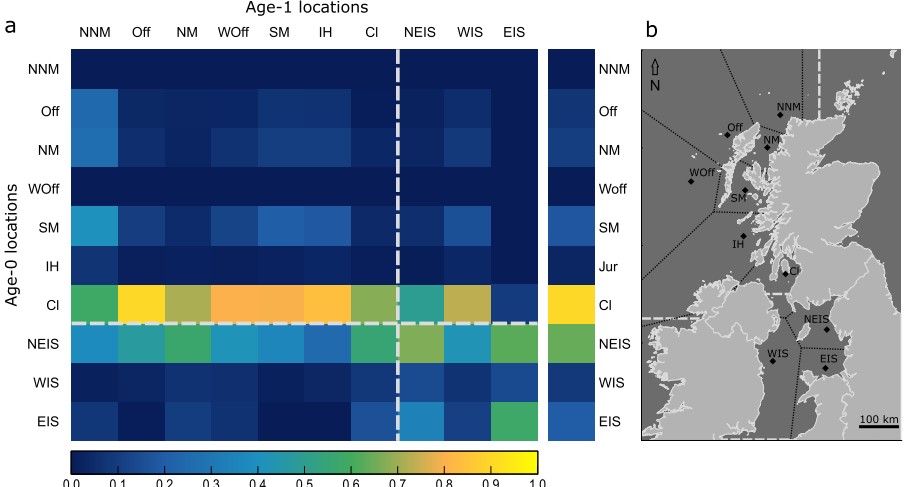

**Fig. 5 Matrix of the likely juvenile origins of mature age-1 whiting caught at ten locations and the contribution of juveniles from these areas to the west of Scotland and Irish Sea. a** Reading down the columns, the matrix displays the likely age-0 origin of age-1 whiting. The matrix rows show the likely contribution of age-0 whiting supplying the same ten areas. The right matrix margin displays the overall contribution to the study area from the ten designated areas. The grey dashed lines indicate the current stock boundary between the West of Scotland—ICES sub area VIa and the Irish Sea—ICES sub area VIIa. **b** The map displays the ten, age-1 whiting catch locations: North-North Minch (NNM), Offshore (Off), North Minch (NM), West Offshore (WOff), South Minch (SM), Inner Hebrides (IH), Firth of Clyde (CI), Northeast Irish Sea (NEIS), West Irish Sea (WIS) and southern East Irish Sea (EIS).

concentrations in the GAMs should be treated with due caution. During model selection, all explanatory environmental variables were rejected from the models. The soap-film smoother constructed from spatial covariates only proved to be a better proxy for the spatial processes governing otolith elemental concentrations than temperature, salinity or the sediment type where fish were caught. The GAMs were used primarily as predictive models and as such, cross-validation showed they performed well. However, the test data used to assess model predictive performance was derived from fish from the same trawl locations as the training data. This may artificially improve the outcome of cross-validation making an independent sample set sourced from a variety of locations much more robust. The likely outcome of cross-validation being conducted on a non-independent test data set is that the model will be inflated and be over fitted to the data. In the absence of a dedicated test set a Monte Carlo cross-validation method was adopted. This technique has been shown to select more parsimonious models than other cross-validation (CV) methods like leave-one-out and decreases the risk of over-fitting during the calibration phase[57]. However, future, work intending to develop chemoscapes of otolith microchemistry should include a dedicated test data set randomly sampled from other locations making the cross-validation more robust.

In the specific context of the West Coast of Scotland, the fishers most likely to impact whiting nursery areas and juvenile aggregations are those targeting *Nephrops* and scallops[58]. As many of these fishers lack quota for whitefish they obtain no benefit from increases to whitefish abundance, but could suffer economic losses if their activities are restricted in order to reduce fish bycatch or protect nursery habitat. In contrast, other fishers targeting whitefish could benefit from larger stocks if nursery areas and juveniles are protected, but because they catch adults in other areas would not be affected by management measures necessary for this protection. The connectivity analysis methods shown here could inform decisions about which nursery and juvenile areas should be protected, and even inform novel mechanisms to share the costs and benefits of nursery area protection between sectors.

Area closures may be a suitable management option for whiting where key life stages are localised, provided they are of a size suitable for management. For example, closing the entire west coast of Scotland would not be practical. However, areas where trawling is prohibited, either permanently or seasonally (e.g., during spawning or after settlement) may allow for renewal of demersal stocks at critical life stages by reducing recruit fishing mortality, enabling their sustainable exploitation in neighbouring areas[59]. The results of this study, a clear demonstration of the movement patterns of whiting recruits, are fundamental data for the design of appropriate closures.

The movements of juvenile to adult stage whiting connect most inshore waters to the west of Scotland and the eastern Irish Sea. These within cohort patterns of movements from aggregated post-settlement, age-0 juveniles in October to age-1 in March are consistent with the distributions predicted by Burns et al.[30]. Mature age-1 whiting occupying waters to the north west of Scotland are most likely to be derived from settled age-0 fish from the Firth of Clyde and the eastern Irish Sea. Age-1 fish occupying waters further south, in the Firth of Clyde and eastern Irish Sea, are most likely recruited from more local age-0 whiting. Local recruitment is less important for western Irish Sea whiting. Like whiting from waters off the west coast of Scotland, age-1 whiting in the western Irish Sea are likely derived from age-0 fish from the Firth of Clyde and eastern Irish Sea. Our results emphasise the importance of the inshore areas of the Scottish west coast to juvenile whiting and their role in contributing to adult stocks. Additionally, Firth of Clyde and east Irish Sea waters play a key role in the recruitment of adult fish to Scottish West Coast waters. Clear connections between the west Irish Sea, the east Irish Sea and the Firth of Clyde are evident here. The northerly movement of Irish Sea fish into the Firth of Clyde indicates a close link between these two areas not accounted for in the current management for this species. These movements have important implications to the current management of whiting.

In the area studied, to the west of the UK, demersal fish stocks are managed and assessed as two separate units: the west of Scotland—ICES sub area VIa, and the Irish Sea—ICES sub area VIIa. However, there is considerable connectivity between the Irish Sea, Firth of Clyde and Scottish west coast, suggesting that these groups are part of a wider whiting population. Furthermore, Tobin et al.[29] demonstrated substantial exchange in juvenile

whiting between the west coast of Scotland and the North Sea, which are also managed as separate units. Seasonal movement and probable migration between these areas could create inaccuracies in stock assessments, especially in stocks where analyses are annually conducted rather than seasonally[17]. Assessment areas that do not properly account for connectivity across boundaries will misinform stock–recruitment relationships and growth and mortality parameters used in stock assessment biasing fisheries management[60,61].

As well as the within-EEZ movements seen here, many species cross international boundaries over the course of their lives. Much has been made of the catching of fish in UK waters by fishers from other EU countries. This issue of the "ownership" of fish is clouded in cases where the juveniles that eventually become the fishable adults originate largely outside the UK Exclusive Economic Zone (EEZ)[62], as for plaice in the southern North Sea[63]. Fishers working in a source location may suffer economic impacts as a result of being required to restrict their activities to avoid easily caught aggregations of juvenile fish[64]. Additionally, the relative importance of different nursery areas may vary over time[65]. A more equitable basis for establishing "zonal attachment" should take into account the role of nursery areas in supplying fisheries through ontogenetic movements[66].

Our approach, using spatially explicit models of otolith microchemistry to predict the likely geographic origin of fish, has the potential to help alleviate a potentially intractable problem in fisheries management; that efforts to protect nursery areas might not benefit the fishers whose activities are restricted to achieve this protection. Quantitative methods of the sort developed here could inform the sharing of the benefits of nursery area protection between those who catch the adult fish and those who incurred costs in avoiding these same fish as juveniles.

## Methods

**Sample collection, age estimation and otolith preparation.** Whiting were collected by bottom trawl from seven sample locations in October 2014 and ten sample locations in March 2015 from across the west coast of the UK (Fig. 1a). 200 individuals were collected from each sample site ($n = 100 > 20$ cm and $n = 100 < 20$ cm) and stored frozen at $-20$ °C. Total Length (TL, to nearest mm) was recorded, sex was determined during dissection and both sagittal otoliths removed using ceramic tipped forceps. A TL stratified random sample (38%) of left sagittal otoliths was used for annual age estimation. The otoliths were set in black resin blocks and 50–100 μm transverse sections taken with a high/low speed saw. The age (in years) at length was estimated using the ICES[67] protocol. Multinomial log linear models were fitted using neural networks in the *nnet* package to derive age-length keys for each sampling time point.

Microstructural analysis on a further sub-sample of otoliths ($n = 26$) was conducted to confirm yearly age estimates and identify the age equivalents of fish at the laser ablation sampling pits. Otoliths were selected for daily age estimation from a length stratified random sample of age-0, age-1 whiting. Otoliths were individually fixed and sectioned to produce fine axial cross-section wafers. These were temporarily fixed to glass slides and ground using a lapping machine with abrasive papers and polished with diamond lapping film until sections were transparent at ~50–70 μm thick. Greyscale images of sagittae were captured under transmitted light using and otolith daily increments were counted from the edge to as close to the primordium as possible. Measurements (±0.01 μm) were made from the centre of the core and from the point at which daily age increments were no longer visible to the ventral edge along the same axis as the ablation transect. The relationship between age in days and radius was modelled using GAMs. Further measurements (±0.01 μm) were made on the post ablated otoliths from the centre of the core to the centre of each ablation pit, following the sampling transect. The optimal model (edf = 1.78, $F = 63.11$, $p < 0.001$, $R^2 = 0.85$) was used to predict total age in days for each otolith and the equivalent fish age/time point at each ablation pit.

The 2014 cohort was identified from annual increment counts and otoliths from age-0 (caught in October 2014) and age-1 (caught in March 2015) were used in the analysis (Table 1). Comparisons of material sampled from within otoliths at two time points from one age class should be avoided due to the potential for confounding ontogenetic influences[7]. Therefore, examining profiles from the otolith core to its edge is unlikely to deliver a reliable method to track the movements of fish because of the difficulties of accounting for ontogenetic effects, as well as those of environmental change. Reliable estimates of connectivity can be inferred by comparing material, which relate to the same point, sampled from

equivalent parts of otoliths from fish belonging to one cohort. Material from close to the edge of otoliths has accreted recently in the life of the fish and can provide a chemical signature at a known geographic location (i.e., the sample trawl sites from October 2014). In this study elemental composition from otolith material ablated near the edge of the age-0 whiting otoliths were used to construct the chemoscapes. The predictive ability of the geolocation models was tested by predicting the catch location of age-0 fish from the element composition of material sampled at otolith edges. The catch location predicted by the model was compared to the true catch location and evaluated by calculating the linear error, precision and accuracy of the assignment. The chemistry of the corresponding age-0 portion of age-1 whiting was then compared with the chemoscape. The juvenile origins of age-1 whiting were inferred by geolocating element samples from internal otolith material accreted during October 2014 on the chemoscape.

Ceramic tipped forceps were used to handle right sagittae under laminar flow conditions and the otoliths were placed in Elga ultra-pure (>18 M cm) water and decontaminated in a series of three sonification steps. All equipment was washed in 10% nitric acid solution before use. The right otolith from each pair was mounted (in a random order to prevent sample batch bias) in Araldite resin (Araldite CY212, Agar Scientific) and sectioned axially to create wafer sections (300 μm), which included the cores.

**Otolith microchemistry analysis.** Element composition was analysed using a PerkinElmer Elan DRCII + ICP-MS (PerkinElmer, Buckinghamshire, UK) equipped with a New Wave Research UP213 laser ablation system (LA-ICP-MS), using helium gas (flow = 0.8 L min$^{-1}$) as the carrier and argon plasma (flow = 0.75 L min$^{-1}$). The presence of 18 element isotopes ($^7$Li, $^{23}$Na, $^{24}$Mg, $^{27}$Al, $^{31}$P, $^{44}$Ca, $^{45}$Sc, $^{47}$Ti, $^{52}$Cr, $^{55}$Mn, $^{60}$Ni, $^{65}$Cu, $^{66}$Zn, $^{85}$Rb, $^{88}$Sr, $^{89}$Y, $^{138}$Ba, $^{208}$Pb) were analysed. Pre-ablation transects (100 μm wide) were run in order to decontaminate the surface before a series of 55 μm diameter pits were sampled for a dwell time of 20 s (~50 μm deep) from the core to the edge of each otolith (6 for age-0 and 12 for age-1). The laser wavelength was 213 nm and pits were ablated at a shot frequency of 10 Hz. At the beginning of each experimental run and after each otolith was sampled, a helium gas blank, NIST and MACS (NIST 612; National Institute of Standards and Technology and MACS 3; United States Geological Survey) ablation were also taken for calibration and instrument drift correction.

Blank subtracted count data were gathered for each ablated pit and converted to element concentrations (ppm) by manual calculation using Ca as the internal standard. The calculated ratios, using the internal standardisation equation[68], compensated for variation in analyte yield between samples and standards. Mean percent relative standard deviations (% RSD) calculated for MACS 3 carbonate standard during the analyses were 8.11 (Li), 7.76 (Na), 25.46 (Mg), 21.40 (Al), 13.88 (P), 15.41 (Sc), 23.64 (Ti), 8.74 (Cr), 12.41 (Mn), 11.64 (Ni), 9.07 (Cu), 206.88 (Zn), 78.81 (Rb), 10.04 (Sr), 13.8 (Y), 18.61 (Ba), 25.7 (Pb). Mean percent relative standard deviations (% RSD) calculated for NIST 612 standard glass were <0.001 (Li), 17.31 (Na), 7.94 (Mg), 12.06 (Al), 13.88 (P), 31.55 (Sc), 15.58 (Ti), 8.62 (Cr), 6.38 (Mn), 5.73 (Ni), 6.12 (Cu), 6.71 (Zn), 78.81 (Rb), 17.83 (Sr), 6.10 (Y), 12.15 (Ba), 8.47 (Pb). All elements are expressed as concentrations. Ablated material from the edges of age-0 otoliths provided element signatures at known geographic locations to produce baseline reference signatures. The geolocation of mature age-1 fish earlier in their life history was inferred from ablated pits corresponding to material laid down in October 2014 at an equivalent time to when the age-0 fish, from the same cohort were caught (Fig. 1b).

**Otolith element concentration and chemoscape modelling.** Maps of environmental variables including temperature, salinity and sediment type were constructed following the method of Burns et al.[30]. The otolith element concentrations were modelled as functions of environmental and spatial variables using Generalised Additive Models (GAMs) in the *mgcv* package. Continuous environmental variables were fitted with penalised cubic regression splines. The spatial variables, longitude and latitude were fitted with a two-dimensional soap-film smoother to prevent "bleed-over" across promontories and peninsulas. Possible collinearity between explanatory variables was explored using variance inflation factors (VIFs) and Pearson correlation coefficients. Combinations of spatial and environmental variables were only considered if VIF values of 3 or lower were achieved and correlation coefficients did not exceed 6. A combined backwards/forward stepwise model selection using AIC and CV was used to select optimal model structures. A random sample stratified across trawl locations and equivalent to a 4:1 data split was used to provide training and test data sets. CV was conducted on 1000 iterations of random training sets and Spearman's rank correlation coefficient used to assess the accuracy of predictions. During model selection the environmental variables were dropped, and every optimal element concentration model contained only the spatial smoothers. All environmental variables were dropped from further analysis. The element models were predicted and compiled as rasters with cells of size 7.4 km ($x$) by 13.3 km ($y$) (Fig. 2).

**Geolocation likelihood assignment from otolith chemoscapes.** The chemoscapes derived from the GAMs of element concentrations provided the mean and variance values, which were used to calculate distribution parameters for each cell. A likelihood estimate was generated for a given data point belonging to the

specified distribution for each cell across the chemoscape. An uninformative prior was assumed that the origin of any individual fish was equally likely across all cells. For elements modelled using Gaussian distributions the mean and standard deviation (estimated from standard error) could be passed directly to the function. The parameters of Gamma distributions (shape and rate) were estimated from the mean and standard error. The sum of (log)ML values for all elements included in a model was calculated for each cell in the landscape. The most probable location for the origin of an individual fish was derived from the highest likelihood value across the model range.

Geolocation model selection was conducted by comparing the predicted catch locations for age-0 whiting with the true sample locations. Five hundred iterations of Monte Carlo cross-validations were run[57], with each iteration randomly sampling a proportion of the data without replacement to make a training set. The remaining data became the test set for that iteration. The random samples were stratified by sample site and the data was split with 60% being used in training sets. This was the lowest ratio split possible given the numbers of fish in some trawl samples (i.e., 5). Backwards stepwise model selection was conducted from a full model containing all six spatially variable element chemoscapes. The accuracy, precision and mean linear error of the predictions were pooled for the 500 cross-validation runs and used to compare models. The optimal model containing Sr and Mn was then further tested in a round of forward stepwise model selection. The model accuracy was defined as the proportion of individuals assigned to a given area (defined by the precision). Precision was defined as the proportion of the posterior probability distribution, which was likely to contain the correct assignment and was thus related to the number of cells (or area) in the landscape[36,69]. The AUC of the accuracy/precision plot was used as a measure for model comparison in combination with the mean linear error (Euclidean distance) between predicted and actual locations. The geolocation models trade-off the precision of predicted locations for the accuracy of that prediction. An optimal precision threshold was derived from the accuracy/precision plots. The threshold ($\theta$) for classifying unknown individuals was considered to be the value, which maximised both precision and accuracy. $\theta$ can be visualised graphically as the longest vertical line separating a diagonal line (with intercept = 0 and slope = 1) and the plotted line of accuracy against precision (Fig. 3).

Predictions of the likely nursery origins of individual mature age-1 fish were possible. The optimal threshold was used in the binary classification of cells as likely or unlikely origin locations. To explore the connectivity of whiting across the Scottish west coast and Irish Sea, for each sample the values of likely locations for each fish were summed and divided by the number of fish in the trawl (as per Trueman et al.[37]). This index, ranging from 0 to 1, was assigned to each cell and the most likely origin locations for fish belonging to each regional group were plotted. A matrix was built to compare local contribution of juvenile whiting from across the Scottish West Coast and Irish Sea to the locations where age-1 whiting were caught. The local areas were defined by defining Voronoi cells (Fig. 5b) using the ten, age-1 catch locations as seeds. The mean proportional recruitment from and contribution to the ten designated areas was then calculated and displayed in a matrix (Fig. 5).

**Statistics and reproducibility**. All statistical and GIS analysis were conducted in R 3.6.3 and details of all data and statistical analyses, including the packages used are presented in the "Methods" section for reproducibility.

**Reporting summary**. Further information on research design is available in the Nature Research Reporting Summary linked to this article.

## Data availability

All of the data used in this publication for figures and supporting the findings presented here are available online from the University of Glasgow Enlighten database (https://doi.org/10.5525/gla.researchdata.1040)[70].

## Code availability

The data analysis and modelling R code developed here is available in a public GitHub repository (https://github.com/NeilMBurns/Element_Isoscape_geolocation20) and has been deposited in the Zendo DOI-minting repository (https://zenodo.org/badge/latestdoi/281963681)[71] and is available from the corresponding author on request.

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

## Acknowledgements

This project was funded by the Scottish Government Clyde 2020 project (ST02i) through Marine Scotland Science. We gratefully acknowledge the officers and crew for conducting the Scottish West Coast Groundfish Surveys (SCOCGFS) and Northern Ireland Ground Fish Survey (NIGFS) who collected the fish samples. We thank Kirsty Donald and Craig Robinson for undertaking the LA-ICP-MS sample analysis.

## Author contributions

N.B. conceptualised the data analysis methods, designed and performed the statistical modelling analysis. N.B. and C.H. wrote the paper. P.W. and D.B. conceived the project idea and reviewed and edited the manuscript.

## Competing interests

The authors declare no competing interests.
