## [Peer Review File · Communications Biology]

Reviewers' Comments:

Reviewer #1:

Remarks to the Author:

Here, Burns et al. use a novel technique in the determination of life stage connectivity for whiting in the Northern British Isles. Specifically, they identify a potentially important nursery (Firth of Clyde), as well as connectivity across demarcated stock areas. The approach that they've used (i.e. continuous pit ablation) is worth investigating further and can be useful to future LA-ICP-MS studies. There are, however, some questions I have about their methods, results and conclusions. Specifically, I have concerns about an approach that assesses all elements in an otolith and then uses them to reconstruct a chronology. As has been shown by Thomas et al. 2020 (MEPS), some elements assessed add noise and limit or confuse data interpretation. Particularly of concern are the transition metals. These elements have been shown to be likely under endogenous control and not reflective of ambient environment. Mn, for example, while it can be linked to exogenic events such as hypoxia, is also associated with proteins involved in Biomineralization (Thomas et al., 2019, Thomas and Swearer 2019).

Overall, this paper presents as a useful contribution to both the fields of otolith microchemistry and fisheries management, and, with amendments listed below, should be published.

Introduction

Line 89 – Please list common name, followed by bracketed species name.

Line 92- Please be consistent, using either "by-catch" or "bycatch".

Line 92 – Nephrops should be italicized.

Results

Table 1 – Please include a space between "0" and "origin".

Line 138 – You assessed for several elements. However, many of these are known to be under strict endogenous control (see Barbee et al., 2014; Thomas et al., 2017). Of particular concern are the transition metals Cu, Zn and Ni. How can you be certain that variation in these elements are due to environmental conditions, rather than individual and cohort physiological differences? These latter three metals almost exclusively act as protein co-factors, with Zn being a known cofactor to key biomineralization proteins (matrix metalloproteinase 2, see Somerville et al. 2003; Visse et al. 2003; Page-McCaw et al., 2003; Thomas et al. 2019; Green et al. 2019; carbonic anhydrase, see Mugiya et al. 1979; Thomas et al., 2019). I have some concerns with this approach, i.e. using multiple elements to reconstruct spatial histories. The inclusion of too many elements to your model may result in adding unnecessary noise to your data set, and the consequent drawing of inappropriate conclusions. The inclusion of Mn is discussed later.

Table 2 – This is a truncated list of elements. If, however, the authors sampled for many elements, but only used the elements listed here, then this should be articulated more clearly in the text. Of the elements listed, I have some concerns with Mg and Na being used as indicators of exogeny. Mg is tightly regulated and exists in high concentrations in fish endolymph. Furthermore, its levels of otolith incorporation are unclear. However, as shown by Limburg et al. (2018, RFSA), it may be an indicator of high metabolic activity. Indeed, Mg is a known interactor with ATP and other hallmarks of growth activity (Fraústo da Silva & Williams 2001). Magnesium's role in otoliths is discussed in Thomas et al. 2020 (MEPS). As for Na, there are few studies illustrating its efficacy in being utilized in spatio-temporal reconstructions.

Line 155 - How do you account for Mn being a known biomineralization cofactor (Tagliabracci et al. 2012, Xiao et al., 2013; Liu et al. 2018) that interacts specifically with the group of biomineralization proteins that Starmaker and its homologs belong to (SIBLINGs)?

If Mn is indicative of exogeny - which it can be (specifically hypoxia; see Limburg et al., 2015; Mohan and Walther 2016; Altenritter et al, 2018) do you have reasons to suppose that these regions (i.e the E Irish Sea) would have higher concentrations of Mn than other regions? However, a laboratory study on Mn's ability to be incorporated into otolith increments (see Mohan et al.

2014) was inconclusive as to its utility as a marker in LA-ICP-MS.

Discussion

While much discussion is given over to the differences between fisheries areas, and the exchange of juveniles between them, it would be good to have some discussion and interpretation of the specific differences in otolith chemistry. Considering that you used otolith microchemistry to infer spawning locations, are there reasons to suppose why one area differs to another in its chemistry? It would be good to have more in the discussion as pertains to the specific differences in elemental contributions to your model.

Methods

Line 371 – You used sonication and 18M water to clean otoliths. How certain are you that you have eliminated the contribution of endolymph elemental concentrations to your analyses?

Line 380 – I would like a lot more information regarding the specifics of the LA-ICP-MS analyses. Specifically; %RSD for each element, spot sizes, flow rates for all gases employed, sampling times, laser frequency/wavelength, mean+/- SD scans per sample, standardization approach of elements.

Line 382 – Sc, Ti, Cr, Mn have typographic errors. Please eliminated the erroneous “24” superscript for each of these elements.

Line 383 – “was analysed” should be “were analysed”

Line 387 - How did you correct for drift? Did you also employ an internal standard? I see you assessed for Yttrium - was this purely as an exogenous indicator, or for reasons of ICP-MS standardization (as it is often used as an instrument standard)?

Reviewer #2:

Remarks to the Author:

The manuscript by Burns et al. uses otolith chemistry methods to assign putative nursery origins to a fished species in the northern Atlantic. The primary novelty of the work appears to be the quantitative method for estimating origins based on a groundtruthed sample. The authors find that the nursery-to-fishery connectivity patterns are complex and cross some existing management boundaries.

Overall I found this work interesting but limited in its scope of interest. The authors are conceptually using a similar approach to quantifying marine population exchange rates (geochemical signatures) that have been applied in a number of other species in various systems throughout the world. The novelty seems to be the continuous surface method. But few details are given about this method, and the authors do not clearly outline what specifically is novel and improved about this method compared to prior assignment methods. Most of the manuscript is oriented towards the specific fishery management case study under investigation. That's fine, but it limits the scope and interest. Right now, the manuscript reads like it would be most suitable for a more specialized fishery journal, such as the ICES Journal of Marine Science. There would be quite a receptive and interested audience in the readership of that type of journal (and I think the outcomes of this work would be more impactful there).

If the authors want to publish this in a broader journal, I think it would need to be significantly reconfigured to focus on the continuous surface method itself, with the whiting just as an example for the application of that method. As an otolith chemistry practitioner myself, I don't come away from the current manuscript with a good sense of how to pursue the continuous surface method in my own work, which I think would be a goal if publishing in a general interest journal (demonstrating how to apply this new method in a fish context).

Specific comments:

Lines 103-105. Starting here could be a place to elaborate on continuous surface approaches, and their differences with classical approaches.

Lines 121-124 Later in the methods you mention 55 μm 'pits'. Is that the diameter of the spot? What is How does that compare to increment widths? How much 'time' is ablated by a spot? How confident are you in assuming fish have not moved in that time period?

Lines 138-140. I did not see assessments of analytical precision or accuracy. Please provide.

Table 2 and Lines 148-150. The table lists 6 elements that were retained, but then the text makes it sound like only Sr and Mn were used for assignments. Please be very clear about which elements were used to assign fish.

Fig 2. Provide units. Are these molar ratios to Ca?

Line 169. Was this a leave-one-out type of validation method?

Lines 191-195. Please clarify, why is it relevant to examine fish origins in October? What ecologically is especially relevant about that month?

Lines 201-203. For semantics, I'm not sure you're really assessing "self-recruitment", which implies they are recruiting in the same location they're spawned. You're looking at linkages between nursery grounds and eventual fishery capture. But that is not natal origins (if they moved prior to settlement in nursery grounds). Be careful with the terminology, and you may want to pick a phrase that is more clearly reflective of what you're measuring (nursery origins).

Lines 243-244. Does this method force-assign fish to the modeled domain only? So there's no mathematical possibility of originating outside the baseline signature space? You may want to comment on whether that matches ecological reality (are there nursery areas outside of your sampling domain?).

Line 384. What was the dwell time and depth of the ablated spots?

Reviewer #3:

Remarks to the Author:

The paper "Connecting fishing grounds to nursery areas using novel otolith isoscape analysis" describes a novel continuous-surface otolith microchemistry approach to estimate the nursery ground origins for a commercially important fish and to estimate connectivity across life stages. Overall I think it's an excellent paper and I'm excited to see how it will catapult this field forward in thinking about new ways to treat and model otolith elemental data. While isotope ratios are becoming more 'standard' a method to be used in movement and connectivity reconstructions, many researchers still only collect elemental data and I feel like the modeling methods we use for element data have been slow to catch up with other approaches in this field. Most element-based assignments have relied on 'forcing' individuals into groups, but this is the first - that I'm aware of - example that has tried using a multi-elemental continuous surface approach. As such this paper represents a methodological leap. But because fish have to keep their blood chemistry in balance, the processes driving element concentrations are typically more complex and debated than isotope ratios, particularly when variations in the ambient water chemistry are small, such as in marine waters (see point 1 below). Given this - I do think the authors need to more clearly and transparently describe the assumptions and potential limitations of this approach, as well as

identify areas requiring further study/ground truthing. I also think the authors could add more testing how well their approach really works for predicting a fish's nursery ground (suggestions below). As it stands, I think they might have overestimated its predictive power - particularly for nursery areas between juvenile sampling locations. I also think - given that the innovation in this study lies in the novel modeling approach - that the authors should seriously consider publishing the code to encourage its use and to increase progress in this field. That all said, I think that this is such a novel approach that I do not think the issues I bring up detract from the overall impact of the paper and I think it marks a new era in otolith chemical analyses.

One thing I kept feeling, particularly in the very opening paragraphs is that - particularly for this journal - the big picture concepts could be broadened to make it more general ecology, less marine fisheries focused. Fish with adult stocks in oceans often have nursery areas concentrated in rivers, deltas, estuaries, nearshore areas that are geographically segregated from harvest grounds. For these, stressors related to human activities such as pollutants, windfarms etc could have a big impact on early life stages. I would recommend adding more about metapopulation dynamics. Use terms like critical habitats and portfolio effects. Borrow language and concepts from Brennan et al 2019 Science. I often think people often "think marine" or "think freshwater". In the latter, folks are constantly talking about shifting habitat mosaics and dynamic riverscapes, but the processes occur in the ocean and you could get habitat patches with higher productivity in a given year due to particular physicochemical patterns (e.g. timing/location of upwelling) or patches with reduced productivity because of spatially explicit stressors (e.g. toxic spills). Collectively (alongside factors like larval drift) these create variation in the productivity and contribution rates of nursery habitats. But some nursery grounds will consistently contribute disproportionately to the adult stocks and those could become our MPA candidates. or be used to support periods/areas for fishery closures.

See attached pdf for detailed comments/queries/suggestions. The main things I would like the authors to think about more:

1) Physiology

I was surprised to see zero mention of physiological effects in otolith elemental tracers. Unlike isotope ratios such as $d_{18}O$ and $^{87}Sr/^{86}Sr$, which have clearly defined mechanisms to relate tissue signatures to ambient environmental conditions, elemental markers have complex uptake, excretion and incorporation mechanisms and thus - in places where chemical variations are typically small (e.g. most marine environments) - it is likely that physiological effects such as growth rate, condition, stress play a larger role. It was interesting the two elements included in the final predictive model (Sr and Mn) have been shown to be modified by both environmental and physiological factors in marine settings - see Sturrock, Thomas and Kalish papers for Sr, and recent Thomas and Limburg papers for Mn (refs below). That said, processes such as upwelling and coastal inputs can still manifest in spatial variation in otolith chemical signatures in the ocean, I just think it is important to acknowledge physiology, particularly as it sounds like environmental covariates were unrelated to the observed otolith signatures. Note that physiological effects can actually work in our favor by augmenting differences among substocks (i.e. the diffs explained by interactions between environment + genotype + phenotype) but they could also complicate these efforts if the training data are not fully comparable to the prediction data (e.g. diffs in sex ratio), or if some phenotypes are more strongly selected against than others between sampling periods (e.g. lose all the slow growing fish). Physiology could also complicate efforts to predict otolith values in unsampled areas - which is effectively what the authors are doing by modeling otolith element concentrations across the entire area using seven juvenile sampling sites. I would be inclined to perform some simple models to test for relationships between [EI] and sex or otolith size-at-age (e.g. otolith radius at in Oct 2014 in the juveniles as you could also reconstruct this in the adults) - maybe using capture loc as a random effect to allow the intercept to vary among areas. If you do find relationships, I wonder if you could account for or remove these effects before performing geolocation predictions. In any case, I don't think this negates the impact of the

paper, but I just want these concepts to be more clearly acknowledged and discussed.

2) Collection sites

Typically, assigning adults to juvenile nursery grounds using otolith chemistry requires (1) cohort matched knowns and unknowns, and (2) all possible sources to be well-sampled (or in this case, modeled). This study does a great job of cohort matching (which is step up from a lot of other studies), but I think it could do a bit more to convince me of the latter. Seven juvenile sampling sites seems like quite a low number, but perhaps they encompass the 7 primary nursery grounds for these adult stocks? It would be good if the authors could give more of a description of how the sampling sites were chosen and overlay juvenile density data on Fig. 1 so we get a sense of sample coverage. And/or – if these 7 sites do not cover all nursery areas - to better demonstrate how this continuous surface approach allows us to predict otolith values in areas without reference samples (see point 3 below). If there were strong relationships between otolith element concs and environmental conditions (e.g. SST, salinity, chl a) I would feel more comfortable as you could use these variables to support the spatial predictions, but I believe the authors did test this and found that adding in environmental covariates decreased rather than increased model fit (again, arguing for the importance of physiology). On that note – it would be great if the authors could more clearly describe that particular effort – it wasn't completely clear how they performed these comparisons (e.g. if they used cross-validation) and exactly what models they compared.

3) Methods and prediction accuracy

In general, I often found it hard to follow the methods and to imagine the model structure used. I was often confused about the exact methods used to define prediction power (in particular L148, 174, 363, 401). It doesn't help that the methods come after the discussion in this particular journal format, but I do feel like there should be more detail in the main text so the reader can get a decent sense of the key steps. For certain parts, even when I read the full methods section I was still unclear. I assume that word limits are pretty strict but given that the big hitter in this study is the novel method, my suggestion is to shift the balance more towards the method and high level concepts, less to the study system specific results and take homes.

Something that I kept being confused about (and I may have just misunderstood it) was that how the authors estimated the accuracy of geolocation predictions and the predictive power of the model (e.g. lines ~174). I couldn't tell if these were always based on comparisons between known capture location vs. in-sample predictions (fitted values) or out-of-sample predictions (cross-validated). Table 2 suggests cross-validation was carried out for the 6 element isoscape spatial models, but it was unclear from the paragraph immediately after Table 2 if you did any cross-validation after that point (e.g. I was really unclear about what model you performed stepwise selection on to obtain an "optimal predictive model" – as I don't think you are talking about the ML function at this point, right?). This part needs additional explanation. In general whenever you talk about *prediction* and how well a model predicts something, you should really be basing that on cross-validation (i.e. predicting an outcome for knowns that were *not used in any steps prior to that point*), otherwise it all gets a bit incestuous and you are really just saying how well the model fits the data in hand.

On a bigger note, I've been thinking about how you could really truly test the predictive power of this method. Even if you just left 1 individual out, fitted the models and then predicted the catch location for that individuals, you still have another ~19 individuals from that same geographic 'cluster' in the training dataset, which would presumably always make it more likely that you would assign that 20th individual to that general area. Ideally you would unlimited juveniles from the entire region so that you could see how well the continuous surface predicts origin for juveniles from nursery areas *between* the 7 sampling areas. Assuming you cannot obtain juveniles from other locations during that same time period, you could instead (and this would be the most conservative possible test of prediction power) would be to exclude *all individuals from 1 of your 7 juvenile sampling sites*, perform all the steps and then assign nursery areas for those individuals. Then put those juveniles back in and perform the same process for site 2 of 7 etc. That

way you can really test how well this continuous surface predicts geolocations for nursery areas with *no juvenile reference samples*. Which is – essentially – the most exciting aspect of this method. I realize that with only 7 juvenile sampling sites to start with, this might not produce the kinds of prediction power one might hope for. But as it stands, I don't see strong evidence demonstrating this method's true predictive power (which is critical if it is going to be actively used in management actions such as fisheries closures) and/or the superiority/validity for using a continuous surface rather than the discrete 'group membership' approaches more traditionally used.

4) Connectivity analysis

L227 - I'm not convinced by the circle approach. It sounds like you drew equally sized circles around each age 1 capture locs and then assigned their juvenile nursery area assignments to those circles based on their otolith chemistry? Whatever shape you end up using, it needs to be shown in Fig. 5b to help the reader interpret the text and Fig. 5a. I am trying to wrap my head around the concept of using the adult capture locations to define connectivity with nursery ground areas. I've always worked with species where the nursery grounds are completely geographically separate from the adults, so it's a strange concept to me. I can see from Fig. 1 that the adults and juvenile areas are not so different, but they are not identical (nor am I sure that your juvenile sampling is fully exhaustive across the entire region), so it feels a bit dangerous to 'force' the juvenile age-0s into areas based on adult catch locations. More generally, defining the circle size seems a bit arbitrary, circles will incorporate variable amounts of land and will also have a bunch of potential juvenile habitat not included (unless the circles are massive and overlapping). I would use voronoi polygons instead of circles and to think more about interpretation of this analysis if areas are defined by adult capture sites (rather than known or modeled distributions of juvenile nursery areas).

Some refs to consider focusing on physiological effects (1-5):

1. Sturrock AM, et al. (2015) Quantifying physiological influences on otolith microchemistry. *Methods in Ecology and Evolution* 6(7):806-816.
2. Sturrock AM, Trueman CN, Darnaude AM, & Hunter E (2012) Can otolith elemental chemistry retrospectively track migrations in fully marine fishes? *J. Fish Biol.* 81(2):766-795.
3. Limburg KE, et al. (2015) In search of the dead zone: Use of otoliths for tracking fish exposure to hypoxia. *Journal of Marine Systems* 141:167-178.
4. Thomas ORB, Ganio K, Roberts BR, & Swearer SE (2017) Trace element-protein interactions in endolymph from the inner ear of fish: implications for environmental reconstructions using fish otolith chemistry. *Metallomics* 9(3):239-249.
5. Hüsey K, et al. (2020) Trace Element Patterns in Otoliths: The Role of Biomineralization. *Reviews in Fisheries Science & Aquaculture*:1-33.

Scotland's Rural College (SRUC)
Peter Wilson Building, King's Buildings
West Mains Road
Edinburgh
29th July 2020

Dear Reviewers,

We are resubmitting our research Article, '*Connecting fishing grounds to nursery areas using novel otolith isoscape analysis*' (tracking number: COMMSBIO-20-0960A) to Communications Biology. The article authors believe we have addressed your useful comments. In line with your recommendations and specific comments we have made changes to the manuscript to widen its appeal. We have also followed your main pieces of advice to more thoroughly describe our methods and discuss physiological and environmental mechanisms which influence otolith microchemistry. We think that in addressing the comments from the three reviewers our research article would make a valuable contribution to Communications Biology and input from the reviewers has improved the article.

We would like to thank the three reviewers for their time and very helpful, constructive comments. We hope that you will find your comments addressed satisfactorily and we have aimed to address all the comments suggested. Please find, below the numbered list of comments and an explanation of how we have addressed each of them (in blue).

Yours sincerely,

Neil Burns (neil.burns@sruc.ac.uk)

corresponding author

Reviewers' Comments	Authors' Response to Reviewers
Reviewer 1	
General Comments Burns et al. use a novel technique in the determination of life stage connectivity for whiting in the Northern British Isles. Specifically, they identify a potentially important nursery (Firth of Clyde), as well as connectivity across demarcated stock areas. The approach that they've used (i.e. continuous pit ablation) is worth investigating further and can be useful to future LA-ICP-MS studies. There are, however, some questions I have about their methods, results and conclusions. Specifically, I have concerns about an approach that assesses all elements in an otolith and then uses them to reconstruct a chronology. As has been shown by Thomas et al. 2020 (MEPS), some elements assessed add noise and limit or confuse data interpretation. Particularly of concern are the transition metals. These elements have been shown to be likely under endogenous control and not reflective of ambient environment. Mn, for example, while it can be linked to exogenic events such as hypoxia, is also associated with proteins involved in Biomineralization (Thomas et al., 2019, Thomas and Swearer 2019). Overall, this paper presents as a useful contribution to both the fields of otolith microchemistry and fisheries management, and, with amendments listed below, should be published.	We think all the points raised here in the remarks to authors from Reviewer 1 are addressed below in the specific reviewer comments.
Specific Comments	
Introduction  Line 89 – Please list common name, followed by bracketed species name. 	Now on line 94 We have added “Norway lobster” as common name, followed by bracketed species name.
 Line 92- Please be consistent, using either “by-catch” or “bycatch”. 	We have checked the manuscript for consistency. We have used “Bycatch” throughout.
 Line 92 – Nephrops should be italicized. 	Now on line 97 We have changed to Nephrops.
Results  Table 1 – Please include a space between “O” and “origin”. 	We have corrected this and now include a space between “O” and “origin”. (line 160 – table 1)

5. Line 138 – You assessed for several elements. However, many of these are known to be under strict endogenous control (see Barbee et al., 2014; Thomas et al., 2017). Of particular concern are the transition metals Cu, Zn and Ni. How can you be certain that variation in these elements are due to environmental conditions, rather than individual and cohort physiological differences? These latter three metals almost exclusively act as protein co-factors, with Zn being a known cofactor to key biomineralization proteins (matrix metalloproteinase 2, see Somerville et al. 2003; Visse et al. 2003; Page-McCaw et al., 2003; Thomas et al. 2019; Green et al. 2019; carbonic anhydrase, see Mugiya et al. 1979; Thomas et al., 2019). I have some concerns with this approach, i.e. using multiple elements to reconstruct spatial histories. The inclusion of too many elements to your model may result in adding unnecessary noise to your data set, and the consequent drawing of inappropriate conclusions. The inclusion of Mn is discussed later.	We have added a paragraph beginning, “A suite of elements...” at line 179 to better explain the reasoning behind the choice of elements. We have amended the wording of line 192, “” to clarify that Cu, Zn and Ni were not included in the final model. We have amended the wording (line 190) to clarify that multiple elements were not used in this part of the analysis.
6. Table 2 – This is a truncated list of elements. If, however, the authors sampled for many elements, but only used the elements listed here, then this should be articulated more clearly in the text. Of the elements listed, I have some concerns with Mg and Na being used as indicators of exogeny. Mg is tightly regulated and exists in high concentrations in fish endolymph. Furthermore, its levels of otolith incorporation are unclear. However, as shown by Limburg et al. (2018, RFS), it may be an indicator of high metabolic activity. Indeed, Mg is a known interactor with ATP and other hallmarks of growth activity (Fraústo da Silva & Williams 2001). Magnesium’s role in otoliths is discussed in Thomas et al. 2020 (MEPS). As for Na, there are few studies illustrating its efficacy in being utilized in spatio-temporal reconstructions.	We agree this could have been made clearer. We have added text at line 220 to better articulate the truncation of the initial list of 17 elements and to better explain that at this stage these element have not been used to predict the geographic positions of the fish. We have reworded the start of the paragraph at line 215 to explain that Mg and Na are not included in the final model and to better describe the ML model selection process.

7. Line 155 - How do you account for Mn being a known biomineralization cofactor (Tagliabracci et al. 2012, Xiao et al., 2013; Liu et al. 2018) that interacts specifically with the group of biomineralization proteins that Starmaker and its homologs belong to (SIBLINGs)? 8. If Mn is indicative of exogeny - which it can be (specifically hypoxia; see Limburg et al., 2015; Mohan and Walther 2016; Altenritter et al, 2018) do you have reasons to suppose that these regions (i.e the E Irish Sea) would have higher concentrations of Mn than other regions? However, a laboratory study on Mn's ability to be incorporated into otolith increments (see Mohan et al. 2014) was inconclusive as to its utility as a marker in LA-ICP-MS.	In response to both comments 7 and 8 where we previously neglected the importance of physiological mechanisms of biomineralization and, importantly for our work, exogeny related to Mn, we have expanded on and developed this comment by adding text at line 193, "The geographic variability of water chemistry for several of these six elements ... "
Discussion 9. While much discussion is given over to the differences between fisheries areas, and the exchange of juveniles between them, it would be good to have some discussion and interpretation of the specific differences in otolith chemistry. Considering that you used otolith microchemistry to infer spawning locations, are there reasons to suppose why one area differs to another in its chemistry? It would be good to have more in the discussion as pertains to the specific differences in elemental contributions to your model.	We agree that this could be expanded upon. We have added two paragraphs to the discussion at line 361 and 371 to discuss the differences in spatial trends of element concentrations, comparing our results with other published articles in the same geographic area, in the same and related species and we offer recommendations for future improvements of similar work.
Methods 10. Line 371 – You used sonication and 18M water to clean otoliths. How certain are you that you have eliminated the contribution of endolymph elemental concentrations to your analyses?	The otoliths were triple rinsed in the 18M water. we have clarified this in line 504. The authors think that given the process, which has been used in several published studies, involves a series of sonification steps following removal, the otoliths were sectioned and the cleaned surface was then pre-bleated we do not think that it is likely for there to be an influence of endolymph chemistry.

11. Line 380 – I would like a lot more information regarding the specifics of the LA-ICP-MS analyses. Specifically; %RSD for each element, spot sizes, flow rates for all gases employed, sampling times, laser frequency/wavelength, mean+/- SD scans per sample, standardization approach of elements.	We have expanded on this section in the methods. We have added the %RSD for all elements measured against MACS and NIST standard glass at line 527. The spot size was originally included in the paragraph at line 380 “55 µm” and has now also been included in the results section where we describe our rationale for the choice of elements. We have also included the word diameter to clarify the pit size. We have added gas flow rates to line 514. Detail has been added to line 516 to 518 describing pit depth and dwell time. Laser details are added to line 519
12. Line 382 – Sc, Ti, Cr, Mn have typographic errors. Please eliminated the erroneous “24” superscript for each of these elements.	We have amended the Sc, Ti, Cr and Mn typographic errors and the erroneous “24” superscript has been deleted (line 515).
13. Line 383 – “was analysed” should be “were analysed”	We have changed to “were analysed” (line 516)
14. Line 387 - How did you correct for drift? Did you also employ an internal standard? I see you assessed for Yttrium - was this purely as an exogenous indicator, or for reasons of ICP-MS standardization (as it is often used as an instrument standard)?	We have reworded the description at line 519 to better describe the internal standards used and correction of instrument drift.
Reviewer 2	
General Comments The manuscript by Burns et al. uses otolith chemistry methods to assign putative nursery origins to a fished species in the northern Atlantic. The primary novelty of the work appears to be the quantitative method for estimating origins based on a groundtruthed sample. The authors find that the nursery-to-fishery connectivity patterns are complex and cross some existing management boundaries. Overall I found this work interesting but limited in its scope of interest. The authors are conceptually using a similar approach to quantifying marine population exchange rates (geochemical signatures) that have been applied in a number of other species in various systems throughout the	We have addressed similar comments made by Reviewer 3 to broaden the readership of the paper, which we feel address this point. We have added and edited the first paragraph of the introduction to relate the concepts in this paper to “big picture” ecology. Additionally, we have now removed the entire paragraph originally L74 –L81 to make the paper less fisheries focused. We think we have addressed the major concerns described here by addressing this reviewer’s specific comments and also in the course of addressing the comments of other reviewers.

world. The novelty seems to be the continuous surface method. But few details are given about this method, and the authors do not clearly outline what specifically is novel and improved about this method compared to prior assignment methods. Most of the manuscript is oriented towards the specific fishery management case study under investigation. That's fine, but it limits the scope and interest. Right now, the manuscript reads like it would be most suitable for a more specialized fishery journal, such as the ICES Journal of Marine Science. There would be quite a receptive and interested audience in the readership of that type of journal (and I think the outcomes of this work would be more impactful there). If the authors want to publish this in a broader journal, I think it would need to be significantly reconfigured to focus on the continuous surface method itself, with the whiting just as an example for the application of that method. As an otolith chemistry practitioner myself, I don't come away from the current manuscript with a good sense of how to pursue the continuous surface method in my own work, which I think would be a goal if publishing in a general interest journal (demonstrating how to apply this new method in a fish context).	
Specific Comments	
1. Lines 103-105. Starting here could be a place to elaborate on continuous surface approaches, and their differences with classical approaches.	We agree that expanding on the discussion of the differences between the two approaches will improve the paper. We have added to the paragraph at line 110 to describe the traditionally used classification method and our proposed continuous assignment approach.
2. Lines 121-124 Later in the methods you mention 55 um 'pits'. Is that the diameter of the spot? What is How does that compare to increment widths? How much 'time' is ablated by a spot? How confident are you in assuming fish have not moved in that time period?	We have added text at line 151 to answer these points, describing the laser diameter and equating this to the increment width and then extrapolating how far potentially a fish could move in that time.

3. Lines 138-140. I did not see assessments of analytical precision or accuracy. Please provide.	We have added detail to the methods section at line 527 to state the %RSD. The authors felt this would be better placed in the methods and is in line with other reviewers' comments.
4. Table 2 and Lines 148-150. The table lists 6 elements that were retained, but then the text makes it sound like only Sr and Mn were used for assignments. Please be very clear about which elements were used to assign fish.	The authors agree that this could have been made clearer. We have added text from line 215 to better explain the selection of which elements to include in the geolocation models and which were finally used to assign fish.
5. Fig 2. Provide units. Are these molar ratios to Ca?	We used blank subtracted count data gathered for each ablated pit and converted to element concentrations (ppm) by manual calculation using Ca as the internal standard as described in the methods. We agree though that this would be useful to know earlier in the paper and have included the units in figure 2 above the legend scale bars.

6. Line 169. Was this a leave-one-out type of validation method?	To assess the predictive ability of the model we did not require to separate the data into test and training sets or use a leave-one-out approach. The predictive ability of the geolocation model was based on assigning individuals (of known catch location) to a predicted catch location. After re-reading this section we feel this is well explained and also offer two references where the method has also been used successfully. However, we are aware that there are many phases to the statistical approach and that the cross-validation used in the previous step may cause confusion. We felt it was best to expand on the GAM cross-validation and have added text at lines 205 -206 to make this distinction clearer.
7. Lines 191-195. Please clarify, why is it relevant to examine fish origins in October? What ecologically is especially relevant about that month?	The authors agree that this point is worth making clear. We have changed the wording in the introduction at line 106 and wording in the discussion at line 410 to state that these are post-settlement fish.
8. Lines 201-203. For semantics, I'm not sure you're really assessing "self-recruitment", which implies they are recruiting in the same location they're spawned. You're looking at linkages between nursery grounds and eventual fishery capture. But that is not natal origins (if they moved prior to settlement in nursery grounds). Be careful with the terminology, and you may want to pick a phrase that is more clearly reflective of what you're measuring (nursery origins).	We agree on the technicality of this definition and have changed our wording throughout, specifically at lines 278, 283, 292, 298, 331 and 335.
9. Lines 243-244. Does this method force-assign fish to the modeled domain only? So there's no mathematical possibility of originating outside the baseline signature space? You may want to comment on whether that matches ecological reality (are there nursery areas outside of your sampling domain?).	The model domain extends several hundred km beyond the age-1 catch locations or is bracketed by land. The model also allows predictions of origin at its extents. While all fish are assigned to a location on the isoscapes we assumed no priors and origins could have been predicted at the edges of the modelled area. Based on previous studies the authors are confident that the age-1 whiting caught at the sample sites shown in figure 5 are likely derived from nursery areas in the model domain.
10. Line 384. What was the dwell time and depth of the ablated spots?	We have added the dwell time and approximate pit depth to line 517

Reviewer 3

General Comments

The paper “Connecting fishing grounds to nursery areas using novel otolith isoscape analysis” describes a novel continuous-surface otolith microchemistry approach to estimate the nursery ground origins for a commercially important fish and to estimate connectivity across life stages. Overall I think it’s an excellent paper and I’m excited to see how it will catapult this field forward in thinking about new ways to treat and model otolith elemental data. While isotope ratios are becoming more ‘standard’ a method to be used in movement and connectivity reconstructions, many researchers still only collect elemental data and I feel like the modeling methods we use for element data have been slow to catch up with other approaches in this field. Most element-based assignments have relied on ‘forcing’ individuals into groups, but this is the first - that I’m aware of – example that has tried using a multi-elemental continuous surface approach. As such this paper represents a methodological leap. But because fish have to keep their blood chemistry in balance, the processes driving element concentrations are typically more complex and debated than isotope ratios, particularly when variations in the ambient water chemistry are small, such as in marine waters (see point 1 below). Given this - I do think the authors need to more clearly and transparently describe the assumptions and potential limitations of this approach, as well as identify areas requiring further study/ground truthing. I also think the authors could add more testing how well their approach really works for predicting a fish’s nursery ground (suggestions below). As it stands, I think they might have overestimated its predictive power - particularly for nursery areas between juvenile sampling locations. I also think – given that the innovation in this study lies in the novel modeling approach – that the authors should seriously consider publishing the code to encourage its use

We have made the 1st paragraph (line 31 on) of the introduction broader in scope to relate ideas of wildlife management to those explored later in the paper. Furthermore we have now removed this entire paragraph originally L74 –L81. This is to address general guidance to make the paper more relevant to a broad readership and less fisheries focused.

The authors think the comments and suggestions raised here under general comments are addressed in our changes and responses to the specific comments below.

and to increase progress in this field. That all said, I think that this is such a novel approach that I do not think the issues I bring up detract from the overall impact of the paper and I think it marks a new era in otolith chemical analyses.

One thing I kept feeling, particularly in the very opening paragraphs is that – particularly for this journal – the big picture concepts could be broadened to make it more general ecology, less marine fisheries focused. Fish with adult stocks in oceans often have nursery areas concentrated in rivers, deltas, estuaries, nearshore areas that are geographically segregated from harvest grounds. For these, stressors related to human activities such as pollutants, windfarms etc could have a big impact on early life stages. I would recommend adding more about metapopulation dynamics. Use terms like critical habitats and portfolio effects. Borrow language and concepts from Brennan et al 2019 Science. I often think people often "think marine" or "think freshwater". In the latter, folks are constantly talking about shifting habitat mosaics and dynamic riverscapes, but the processes occur in the ocean and you could get habitat patches with higher productivity in a given year due to particular physicochemical patterns (e.g. timing/location of upwelling) or patches with reduced productivity because of spatially explicit stressors (e.g. toxic spills). Collectively (alongside factors like larval drift) these create variation in the productivity and contribution rates of nursery habitats. But some nursery grounds will consistently contribute disproportionately to the adult stocks and those could become our MPA candidates or be used to support periods/areas for fishery closures.

See attached pdf for detailed comments/queries/suggestions. The main things I would like the authors to think about more:

1) Physiology

I was surprised to see zero mention of physiological effects in otolith elemental tracers. Unlike isotope ratios such as $\delta^{18}\text{O}$ and $^{87}\text{Sr}/^{86}\text{Sr}$, which have clearly defined mechanisms to relate tissue signatures to ambient environmental conditions, elemental markers have complex uptake, excretion and incorporation mechanisms and thus – in places where chemical variations are typically small (e.g. most marine environments) – it is likely that physiological effects such as growth rate, condition, stress play a larger role. It was interesting the two elements included in the final predictive model (Sr and Mn) have been shown to be modified by both environmental and physiological factors in marine settings – see Sturrock, Thomas and Kalish papers for Sr, and recent Thomas and Limburg papers for Mn (refs below). That said, processes such as upwelling and coastal inputs can still manifest in spatial variation in otolith chemical signatures in the ocean, I just think it is important to acknowledge physiology, particularly as it sounds like environmental covariates were unrelated to the observed otolith signatures. Note that physiological effects can actually work in our favor by augmenting differences among substocks (i.e. the diffs explained by interactions between environment + genotype + phenotype) but they could also complicate these efforts if the training data are not fully comparable to the prediction data (e.g. diffs in sex ratio), or if some phenotypes are more strongly selected against than others between sampling periods (e.g. lose all the slow growing fish). Physiology could also complicate efforts to predict otolith values in unsampled areas – which is effectively what the authors are doing by modeling otolith element concentrations across the entire area using seven juvenile sampling sites. I would be inclined to perform some simple models to test for relationships between [EI] and sex

or otolith size-at-age (e.g. otolith radius at in Oct 2014 in the juveniles as you could also reconstruct this in the adults) - maybe using capture loc as a random effect to allow the intercept to vary among areas. If you do find relationships, I wonder if you could account for or remove these effects before performing geolocation predictions. In any case, I don't think this negates the impact of the paper, but I just want these concepts to be more clearly acknowledged and discussed.

2) Collection sites

Typically, assigning adults to juvenile nursery grounds using otolith chemistry requires (1) cohort matched knowns and unknowns, and (2) all possible sources to be well-sampled (or in this case, modeled). This study does a great job of cohort matching (which is step up from a lot of other studies), but I think it could do a bit more to convince me of the latter. Seven juvenile sampling sites seems like quite a low number, but perhaps they encompass the 7 primary nursery grounds for these adult stocks? It would be good if the authors could give more of a description of how the sampling sites were chosen and overlay juvenile density data on Fig. 1 so we get a sense of sample coverage. And/or – if these 7 sites do not cover all nursery areas - to better demonstrate how this continuous surface approach allows us to predict otolith values in areas without reference samples (see point 3 below). If there were strong relationships between otolith element concs and environmental conditions (e.g. SST, salinity, chl a) I would feel more comfortable as you could use these variables to support the spatial predictions, but I believe the authors did test this and found that adding in environmental covariates decreased rather than increased model fit (again, arguing for the importance of physiology). On that note – it would be great if the authors could more clearly describe that

particular effort – it wasn't completely clear how they performed these comparisons (e.g. if they used cross-validation) and exactly what models they compared.

3) Methods and prediction accuracy

In general, I often found it hard to follow the methods and to imagine the model structure used. I was often confused about the exact methods used to define prediction power (in particular L148, 174, 363, 401). It doesn't help that the methods come after the discussion in this particular journal format, but I do feel like there should be more detail in the main text so the reader can get a decent sense of the key steps. For certain parts, even when I read the full methods section I was still unclear. I assume that word limits are pretty strict but given that the big hitter in this study is the novel method, my suggestion is to shift the balance more towards the method and high level concepts, less to the study system specific results and take homes.

Something that I kept being confused about (and I may have just misunderstood it) was that how the authors estimated the accuracy of geolocation predictions and the predictive power of the model (e.g. lines ~174). I couldn't tell if these were always based on comparisons between known capture location vs. in-sample predictions (fitted values) or out-of-sample predictions (cross-validated). Table 2 suggests cross-validation was carried out for the 6 element isoscape spatial models, but it was unclear from the paragraph immediately after Table 2 if you did any cross-validation after that point (e.g. I was really unclear about what model you performed stepwise selection on to obtain an "optimal predictive model" – as I don't think you are talking about the ML function at this point, right?). This part needs additional explanation. In general whenever you talk about

prediction and how well a model predicts something, you should really be basing that on cross-validation (i.e. predicting an outcome for knowns that were *not used in any steps prior to that point*), otherwise it all gets a bit incestuous and you are really just saying how well the model fits the data in hand.

On a bigger note, I've been thinking about how you could really truly test the predictive power of this method. Even if you just left 1 individual out, fitted the models and then predicted the catch location for that individual, you still have another ~19 individuals from that same geographic 'cluster' in the training dataset, which would presumably always make it more likely that you would assign that 20th individual to that general area. Ideally you would unlimited juveniles from the entire region so that you could see how well the continuous surface predicts origin for juveniles from nursery areas *between* the 7 sampling areas. Assuming you cannot obtain juveniles from other locations during that same time period, you could instead (and this would be the most conservative possible test of prediction power) would be to exclude *all individuals from 1 of your 7 juvenile sampling sites*, perform all the steps and then assign nursery areas for those individuals. Then put those juveniles back in and perform the same process for site 2 of 7 etc. That way you can really test how well this continuous surface predicts geolocations for nursery areas with *no juvenile reference samples*. Which is – essentially – the most exciting aspect of this method. I realize that with only 7 juvenile sampling sites to start with, this might not produce the kinds of prediction power one might hope for. But as it stands, I don't see strong evidence demonstrating this method's true predictive power (which is critical if it is going to be actively used in management actions such as fisheries closures) and/or the superiority/validity for using a continuous surface rather than the discrete 'group membership'

approaches more traditionally used.

4) Connectivity analysis

L227 - I'm not convinced by the circle approach. It sounds like you drew equally sized circles around each age 1 capture locs and then assigned their juvenile nursery area assignments to those circles based on their otolith chemistry? Whatever shape you end up using, it needs to be shown in Fig. 5b to help the reader interpret the text and Fig. 5a. I am trying to wrap my head around the concept of using the adult capture locations to define connectivity with nursery ground areas. I've always worked with species where the nursery grounds are completely geographically separate from the adults, so it's a strange concept to me. I can see from Fig. 1 that the adults and juvenile areas are not so different, but they are not identical (nor am I sure that your juvenile sampling is fully exhaustive across the entire region), so it feels a bit dangerous to 'force' the juvenile age-0s into areas based on adult catch locations. More generally, defining the circle size seems a bit arbitrary, circles will incorporate variable amounts of land and will also have a bunch of potential juvenile habitat not included (unless the circles are massive and overlapping). I would use voronoi polygons instead of circles and to think more about interpretation of this analysis if areas are defined by adult capture sites (rather than known or modeled distributions of juvenile nursery areas).

Some refs to consider focusing on physiological effects (1-5):

1. Sturrock AM, et al. (2015) Quantifying physiological influences on otolith microchemistry. *Methods in Ecology and Evolution* 6(7):806-816.

2. Sturrock AM, Trueman CN, Darnaude AM, & Hunter E (2012) Can otolith elemental chemistry retrospectively track migrations in fully marine fishes? J. Fish Biol. 81(2):766-795. 3. Limburg KE, et al. (2015) In search of the dead zone: Use of otoliths for tracking fish exposure to hypoxia. Journal of Marine Systems 141:167-178. 4. Thomas ORB, Ganio K, Roberts BR, & Swearer SE (2017) Trace element–protein interactions in endolymph from the inner ear of fish: implications for environmental reconstructions using fish otolith chemistry. Metallomics 9(3):239-249. 5. Hüsey K, et al. (2020) Trace Element Patterns in Otoliths: The Role of Biomineralization. Reviews in Fisheries Science & Aquaculture:1-33.	
Specific Comments	
1. L46 change catching locations to catch locations or harvest areas	We have changed to catch locations now in line 55
2. L47 Go big picture ecology, less fisheries focused. The word undersized just makes me think fisheries	We have removed undersized mortality and replaced it with a broader description now in lines 56 - 57
3. L52 – 53 Again broad readership – this is a bit too fisheries centric	We appreciate that this sentence may have been too fisheries specific but think that it makes an important point and is critical to the narrative. We have changed the wording to describe our point in broader language. (line 61 -62)
4. L57 Change to from which recruits originate	We have changed to “from which recruits originate” (line 67)
5. L64 Physiological effects should be mentioned as they can have a significant effect, especially in fully marine fish (see Sturrock 2012, 2014 & 2015, Walther et al 2010 and the recent work by Karin Limburg and Oliver Thomas). For example there is good evidence to suggest growth rate affects elemental uptake, processing and incorporation into the otolith. The good news is that these physiological effects can actually augment among-site differences	We have added text at line 74 to discuss physiological effects on otolith element concentrations. This was an omission. “. Physiological influences have also been shown to effect concentrations of various elements in endolymph and otoliths21–24. The combined effects of physiology and environment on otolith microchemistry may augment among-site differences and result in detectable spatial variation23”

in elemental fingerprints. However, they also mean that the most robust application of otolith elemental concs requires sufficiently large and representative sample sizes and cohort-matching between the juvenile and (juvenile portions of) adult tissues. This study is great that it uses cohort-matching, however it is unclear how representative the juvenile sampling is of all potential nursery grounds, and thus I'm slightly nervous about smoothing the observed values among areas. But it's a massive jump from the clunky DFA analyses that have dominated the otolith chemistry literature till now so I absolutely think it's still an amazing methodological leap!	We address the comments in relation to coverage of juvenile sources below.
6. L65 This does not fit here – otolith centric	We have removed this sentence and added text to link the remaining sentences better on line 79
7. L71 Confirm in sentence if this was based on otolith chemistry. If it wasn't an otolith-based decision, this paragraph should be reworked otherwise it's a mixture of two quite large and important topics (1) otolith element markers and otolith-based management, vs (2) general consideration of life stage connectivity in fisheries management.	We agree that our argument here was not clear. We have altered the text to make our point more clearly, that these are some of only a few examples where otolith research as influenced management. (now lines 82 -86). We have also changed the wording here heeding the general advice to cater towards the broader readership.
8. L76 Strange statement/don't follow logic. Don't all marine fish have the ability to move large distances during their life cycle?	Our language here was confusing. We have now removed this entire paragraph originally L74 –L81. This is to address general guidance to make the paper more relevant to a broad readership and less fisheries focused.
9. L89 Broad readership - help them out! "lobster (Nephrops...)"	We have changed to Norway lobster (Nephrops norvegicus) (line 94)
10. L92 italics and probably just Nephrops now so the same as later in discussion	We have changed to Nephrops (line 97)
11. L93 I would change to 'multiple species'	We have changed to multiple (line 98)

12. L96 I like all these points but I think you just need to drive home the point that some patches are consistently more productive than others and thus the power of your work is to identify those critical habitats	We have added a sentence to lines 100-102 to highlight this point.
13. L104 Very cool! I love the fact this is a new approach but I do think you need to emphasise more that it's novel and awesome (and thus belongs to be in this journal) but that - given the nuances associated with elemental transport - that more ground truthing needs to be done to see whether smoothing across space is really a valid treatment of element conc data	We have emphasised our novel approach by changing the text at line 123 and including an additional sentence at line 125, "Here, we develop a continuous surface assignment approach applied to otolith microchemistry for the first time." We have also added a further explanation to make clear that microchemistry analysis does not have as defined mechanisms for incorporation in tissue as stable isotopes at line 126
14. L109 Not clear by what is meant by 'previously found to be useful'. Also, re-emphasize be clear that you're talking about element concs not isotope ratios, which are more typically used to build isoscapes... you could consider coining a new descriptor like 'chemoscape' but I personally like the term isoscape... I would go with something like 'We modeled the spatial distribution of element concentrations ('isoscapes') measured in the otoliths of juvenile age-0 whiting.'	We agree the wording was clunky and caused confusion. The suggested sentence has been used instead at line 132 We agree that isoscape, although used predominantly in reference to stable isotopes is the better term and hopefully will be adopted to refer to surfaces developed from microchemistry analysis in the future.
15. L113 Go a bit bigger and broader! 'These results reveal significant connectivity across current stock boundaries and suggest that the Firth of Clyde as a critical nursery area supplying juveniles to all major fisheries'. Or something.	We have replaced the highlighted sentence with the suggested sentence at line 136. We refine "...major fisheries" though in favour of "...fished aggregations in two stock areas"
16. L123 I am assuming that you used the same location (growth axis and age) on all otoliths? This is important and should be confirmed here, as different otolith axes grow at different speeds, and elements and proteins are not incorporated homogeneously around the otolith edge. With elemental markers we need to standardize everything way more than isotope ratios	We have added text to clarify this on line 149 and made reference to the image in figure 1. As per the comment below Fig 1 has also been adapted to make it more professional looking.

17. Table 1. That seems strange to me - where did the females go? Is there sex-selective mortality going on? Can you add a brief comment to hypothesize this difference in sex ratio?	We have added a comment as a table footnote to hypothesize the difference in sex ratio at line 163.
18. Table 1. Add space between “0” and “origin”.	We have now included the space between “0” and “origin”. (line 160 – table 1)
19. Figure 1. I love this paper but I feel that this figure could be more polished and professional looking. On a more scientific level - which spot(s) did you use to train model and assign Age1 fish with? Highlight these spots on this image. Assuming you didn't use daily increments to time match among samples and/or if you averaged all spots across that 7 month period (mar-oct). I would have considered larger and/or more frequent spots to avoid within-year temporal mismatching and also to increase detection ability for lower level elements... as it is, there is a lot of otolith material that is *not* sampled between each spots making it even more important to carefully temporally match among individuals and the two year classes.	We have tried to improve the professional look of the figure, also keeping it in line with other similar figures in Communications Biology. We have referred to the spots used to train the model and assign age-1 fish in the figure caption at line 174. Having tried to add a marker or highlight these spots the figure looked too cluttered so we fell it is best to refer to this in the figure caption. These results were part of a larger study. The size and number of spots was selected to balance sample run time and maximise the number of data points. Daily increments were used to verify age readings in years (again part of the wider work). While spot locations on the laser ablation platform were placed by estimation, at these two time points and in these relatively young ages the accretion of translucent and opaque material is fairly obvious. A daily age ~ otolith growth model allowed estimation of the age of each fish at the sample pit and subsequently, the best estimates for spot placement during laser programming were then verified by measuring from core to the ablated pit. Only otoliths with sample pits corresponding to the timepoints of interest were included in the study. We have added this detail to the methods section also from line 469. To improve Fig. 1 we have removed colour on the dashed lines and changed the map shading. We also changed the symbols and colours of the symbols indicating the trawl sample locations.

20. L140 "showed spatial variation based on "x method and y criteria". I know it's hard to include just enough methods to get by in these papers, but you do need at least high level reference to the method used to make that statement as it resulted in a significant change to the number of elements considered in the next part.	We have expanded this section to include more methodology and, specifically lines 190 – 191 addresses this point
21. Table 2. This is interesting to me - Mg, Na are likely be almost an entirely "physiological signal" and Sr and Mn almost certainly have quite a bit of "physiology" in there - see Sturrock et al 2012, 2014 and 2015, Limburg 2018 and 2019, and Oliver Thomas 2018 and 2019 papers. But because you've cohort matched this shouldn't matter too much. I would be interested for you to explore the factors explaining these (e.g. using a lmer with otolith [EI] as the response, with average capture temperature + salinity + sex + otolith radius at capture + (1 site) as predictors just to explore the underlying mechanisms)... Were there significant differences in juvenile size or condition among sites? It would be interesting to	As discussed above these results form part of a wider study. The mixed effects models suggested by this reviewer were explored. However, I (the first author) did not think to incorporate the ideas suggested here re density. Previously published work on ontogenetic movements in this species would allow density per area to be incorporated. The authors (principle the corresponding and first author) would be keen to develop these ideas as per the comment (23) below. We think this may be more useful developed as a new study and would be keen to explore this further.

see if the juveniles from faster growing areas contributed disproportionately to the adult stock (either in absolute terms or normalized to an approx juvenile density per area if those data were available).	
22. L149 Recommend rewording and clarifying this part, and maybe adding a subheader? As it stands, it's confusing having this immediately below Table 2 where you have 6 separate models that basically create 6 elemental isoscapes. I've re-read the main methods and I'm still unclear about what model you performed stepwise selection on and thus how you went from 6 GAMs to a single "optimal predictive model" using 2 elements (in combination?). Also, state how you defined 'optimal' here (AICc? Deviance explained?) and define exactly what you mean by "model prediction". You haven't mentioned cross-validation here, which to me implies you're talking about model fit (diff between observed and fitted)?	To make this section clearer we have added text from line 215, hopefully also better introduce the section and explain "model prediction". We have added a subheading at line 213 for this section and a definition of "optimal" at line 225 To aim to limit repetition we ended up splitting the methods between the results section and the methods section. This obviously has not helped with clarity. We have therefore also added text at line 565, 569 and 573 in the methods section to improve our explanation
23. L153 Given that all your sites are marine and presumably >30ppt I dont think we would see a salinity effect. So my conceptual model is that cold water  slower growth  higher Sr concs and more productive areas  higher growth and otolith protein concs  higher Mn concs (or hypoxic areas  higher Mn concs but I'm guessing none of these areas are ever hypoxic)? No need to add any of this in here I'm just mulling - if you have data that support or refute this please let me know!	As you say all areas are > 30ppt salinity, as you move offshore it reduces slightly (very slightly) at the depths whiting inhabit. Temperature varies between ~9 and 13 °C in the bottom waters getting colder offshore (October) but this can become much more uniform in March. A number of Scottish studies have found high Mn in several gadoid species in the Firth of Clyde. This has been assumed in the past to correlate with high Mn concentration in the water here. These studies did not sample fish from the Irish Sea though for comparison. We collected data from otoliths from the 2013 cohort also and plan to investigate this further. Once a publication decision is made on the current manuscript I (corresponding author) would be keen to discuss this further.
24. Figure 2. This decreasing error at the edge seems strange - I would expect uncertainty to increase the further away from the	We have added labels to the figure and specified units in the caption. The smoother knots are defined at regular spatial intervals within the extent of

observations you get. Could this be something to do with the locations of the knots you defined in the soap film smoother? Rather than using SE you could simulate from the model posterior to create and map prediction intervals. If they allow you to add more labels to the plots please do! e.g. 'Mean Sr (ppb)', SE Sr (ppb) and so on.. If you dont do this you should at least add units to the caption. And I would also remind reader in caption that these are otolith concentrations from age-0 whiting

the smoother. It is defined by a boundary. The boundary can either be specified as fixed or allowed to vary. We opted for a variable boundary (viewed in 2 dimensions this would look like a conventional GAM). A fixed boundary would be suitable for modelling values which approach a constant at the model extents. While the variable boundary will better in this case it may still not portray a realistic picture of the variance.

We have added labels to Fig 2 legend scale bars

25. L169 spell it out to the reader - when we see Fig. 2 where the two elements are shown separately it's a mind-jump to realize you are now talking about combining them both, so just remind the reader here that the ML function uses both the Sr and Mn isoscapes combined.	This is now better explained at the beginning of the section, "Maximum likelihood geolocation model selection" from line 215 We have also added clarification at line 228 and 246 to reaffirm the combination of Sr and Mn in the likelihood model.
26. This isn't truly predictive ability if the same sample was used to fit the model or if it's not a sample from somewhere random across the continuous surface (see main comments in word doc)	We agree a more robust approach would have been to include samples from locations other than those used in the modelling process. While, the geolocation models were fitted on training data and test data generated from the one set of sample locations we felt this optimised the use of available data. In a study of this type there will always be a trade-off between maximising sample size and spreading samples (and indeed test set data). By using GAMs to quantify the error associated with the sample locations we needed to maximise the number of individual fish in each sample. We have clarified this step in the methods at line 574 and made reference to the this in the results at line 219
27. L175 Again - I am not sure here if your predictions are true predictions using cross-validation here (i.e. fitting model without a sample or group of samples in it, and then using that model to predict location of said sample or group of samples) vs. predicting locations of the training data used to fit the model. If the latter, this doesn't actually tell you how well your model predicts nursery location. It gives you a sense of how well your model fits the training data but not its true prediction power.	The response to comment number 26 is also applicable to this comment.
28. L202 It doesn't really work you talking about specific geographic locations by name but to have no locations described on any of the maps in your figures prior to this point. I realize you don't want to make the figures too busy but either the text needs to be broadened to country level (e.g. northwest Scotland) so that a reader can understand the results from Fig. 1 and 4, or somehow	The use of place names in this section is indeed cause for confusion. We have considered the suggestion above and think that keeping the two sections separate as this probably works best and keeping the map within figure 5 is also useful to show (as is suggested below) the stock boundary. Our solution has been to identify the age-1 haul locations in the caption of figure 4 (tried adding them to the figures as titles and as labels but it was

you need to label key place names on Fig. 4. I actually think this section should be incorporated with the next paragraph, and to discuss Fig 4 and 5 together (with the acronyms used in Fig 5 added to Fig 4 to help the reader)

too cluttered) and to rationalise the names used in the section highlighted in this comment to be the same as the names/acronyms used in the following section. We have added a map with the trawl location names to figure 4. From line 275 on all haul sample area names used are the same throughout the manuscript.

We have added a panel to the Fig 4. to specifically name the sample trawl sites and changed the orientation of the legend scale bar to fit the new figure design.

29. L228 See main comments in word doc

“L227 - I'm not convinced by the circle approach. It sounds like you drew equally sized circles around each age 1 capture locs and then assigned their juvenile nursery area assignments to those circles based on their otolith chemistry? Whatever shape you end up using...”

To clarify these points, we have altered text at line 310. We have also taken on board the helpful suggestion of using Voronoi polygons instead of circles and included these in figure 5. We have altered the methods text at line 600 to reflect this also (changed text at line 310 in results in line with these changes). While the resulting matrix does not differ greatly from the one generated by the circle approach, conceptually the polygons are much better to explain.

This species does not have well defined juvenile locations, although they do tend to occupy waters closer to shore than the adults. However, mixing between adults and juvenile shoals is common and you often find immature and mature fish together. Our use of the adult catch locations was based on the fact that this is where they were caught and our general question has been, “where did the fish caught at these locations originate from?” We have included text at line 146 to describe and reference previous work on modelling the distribution of this species and have indicated where the juveniles from that study were most likely to be found.

Fig. 5 has a re-worked map to display the ICES stock management areas and the Voronoi polygons. The matrix is now built from the Voronoi polygons rather than the circles we previously used. NB the stripy effect on the scale bar is only visible in image format as is not seen on the .pdf version which will be used for final submission.

	a  b 30. L239 This also needs to be shown on the map in b	We have added the stock boundary to the map and identified the ides areas in the figure caption.
31. L403 This section needs more detail. It sounds like you tried spatial smoother vs. environmental model, but why not use $s(\text{lat}, \text{long}) + s(\text{Salinity}) + s(\text{Temp})$? You may have done but the language suggests otherwise. Also 'best predictive models' suggests predictions and yet it's not clear if you have done cross-validation	We have added a more detailed explanation to this section (line 542). We had previously kept the description short to avoid over complicating the methods description but now with the inclusion earlier of more detailed discussion of environmental effects on otolith microchemistry more detail here is clearly a better choice.
32. L406 This bit seems clearer (I am assuming you used $Sr \sim s(\text{lat}, \text{long})$ as model formula?) - but in general given how useful this work could be I would request adding sharing your code (github or some other repository)	We will gratefully take this advice (see fuller response to comment 38 below) and have added the code to github
33. L408 help the reader by defining cell size (e.g. 10km resolution or 9.99...)	We have altered this to defined cell size rather than the area (line 558)

34. L412 This part needs to be more clearly worded (and also I don't see where the stepwise model selection that resulted in only Sr and Mn being considered is explained)	We agree, re-reading this section, the wording was not clear. We have changed the wording at lines 562 to 565 and added the stepwise model selection explanation at line 574
35. L415 all cells or all cells in the model? Were some cells excluded from different models?	We have clarified this by removing "in the model." Specifying "in the model" was superfluous (line 566).
36. L442 See above - could you try voronoi polygons instead?	We have addressed this point at comment 29. Briefly though, we have used your suggestion and adopted Voronoi polygons instead of circles.
37. L449 I would put on dryad and github so folks can more readily apply your methods to new datasets/systems. I'm sure it will lead to more citations and collaborations in the long run.	We are grateful for the advice and have uploaded the code to github and the data is now accessible through the University of Glasgow Enlighten database.

Reviewers' Comments:

Reviewer #1:

Remarks to the Author:

Dear Authors,

Thank you so much for addressing my questions regarding your manuscript. Well done on adjusting it, and answering my various queries!

I have no further concerns.

Reviewer #2:

Remarks to the Author:

The authors have done a very good job revising their manuscript. It is significantly improved. I have no further major suggestions, and I believe this can be published.

One tiny suggestion: consider the term "chemoscape" instead of "isoscape". This is broader and more apt to the current data. It is less frequently used, but see Walther & Nims 2015 Estuaries & Coasts.

Nice job. One general thought: this paper is a nice example of how practitioners who analyze a broad suite of elements (AKA the "Kitchen Sink" approach) may likely find that the vast majority are useless for geographic assignment purposes. That's one of my primary takeaways from this current work.

Reviewer #3:

Remarks to the Author:

Excellent work by the authors to address the comments of all three reviewers. This iteration has been greatly improved and I believe it to be suitable for publication. As I said in my previous review, this is a novel application of a traditional 'isoscape' approach to assess fish connectivity, using otolith element concentrations (instead of the more traditional isotope ratios). Given that the costs associated with analyzing element concentrations in otoliths is typically much lower than the costs associated with measuring otolith isotope ratios, they are often the 'tracer of choice', yet the methods used to analyze the resulting data have not evolved as quickly as those used in isotope movement ecology. As such this paper will have a major influence in this field and provide fisheries and ecosystem managers with new tools and approaches to help protect marine and coastal resources.

A few minor remaining points:

L73 = affect

L74 = I think you should be fully candid here and err more on the side of caution. Physiological effects could augment stock-specific differences (e.g. fast growing stocks vs. slow growing stocks) but they could also confound results (e.g. fast growing individuals within a stock being misclassified to a geographically distant fast-growing stock), particularly if every source hasn't been extensively sampled. I suggest modifying the next couple of sentences to something like..
"Physiological influences have also been shown to affect element concentrations in the endolymph and otoliths, potentially complicating their use to recover geographic or environmental information. However, so long as all sources have been adequately sampled and samples are cohort matched to avoid potentially confounding effects of interannual variability, the combined effects of physiology and environment may augment among-site differences in otolith microchemistry and thus increase the power of this tool to reconstruct connectivity patterns. Specifically, elemental

concentrations....”

L91 – the wording sounds a bit strange – would this work instead?

“...has operated further offshore since the 1990s following declines in gadoid abundance.”

L217 – can you help the reader (e.g. use a non-specialist term here)? I had never heard of a “custom dictionary” in this context.

L218-219 and 243-247 - Here and in the methods section, it is still unclear to me if you used independent samples to generate the accuracy estimates provided in Lines 243-247. Those values (e.g. “242 km between the predicted catch locations and the true catch locations”) are what the reader will think the accuracy of your adult assignments are. But I believe they will overestimate model prediction power if they were generated using in-sample predictions (fitted values) instead of only out-of-sample predictions. It sounds like you did split the juvenile data and use Monte Carlo cross-validation for model selection (which is great - Line 567-572). But I couldn't quite work out if you continued to use the split sample when comparing predicted vs known catch locations to assess the final model's predictive power (i.e. fitted values from the 60% of juvenile samples used to train the model vs. predicted values from the 40% remaining 'independent' samples)? Please clarify clearly here and in the Methods. If you did include fitted values I would caveat the accuracy estimates provided in L243-247 as potentially over optimistic if directly applied to the adult assignments.

Ideally (as you point out in the discussion – thanks for adding this) – you would have an incredibly extensive juvenile reference library and leave entire sites out from the model fitting procedure to use as independent test samples. You could perform multiple model fitting exercises subsampling individuals and sites to (1) assess whether using a continuous surface approach is totally valid for physiologically sensitive tracers such as otolith element concentrations and/or (2) whether these tracers simply require a more extensive reference libraries than the more traditional isoscapes (where isotope ratios vary predictably with environmental covariates). But this is all food for thought and will hopefully become your follow up study!

Scotland's Rural College (SRUC)
Peter Wilson Building, King's Buildings
West Mains Road
Edinburgh
16th October 2020

Dear Reviewers,

We are resubmitting our research Article, now re-titled following editorial and your suggestions '*Otolith chemoscape analysis in whiting links fishing grounds to nursery areas*' (tracking number: COMMSBIO-20-0960A) to Communications Biology. We believe we have addressed your final helpful comments. There has been a substantial improvement to our manuscript directly resulting from the constructive comments from this and the previous round of reviews. We think that our research article will now make a valuable contribution to Communications Biology. We are very grateful to all three reviewers for their thorough review and the substantial time they have invested in our work. Please find, below the numbered list of comments and an explanation of how we have addressed each of them (in blue).

Yours sincerely,

Neil Burns (neil.burns@sruc.ac.uk)
corresponding author

Reviewers' Comments	Authors' Response to Reviewers
Reviewer 1	
General Comments Dear Authors, Thank you so much for addressing my questions regarding your manuscript. Well done on adjusting it, and answering my various queries! I have no further concerns.	We very much appreciate the contribution made by Reviewer 1 to improving the quality of our paper.
Reviewer 2	
General Comments The authors have done a very good job revising their manuscript. It is significantly improved. I have no further major suggestions, and I believe this can be published. One tiny suggestion: consider the term "chemoscape" instead of "isoscape". This is broader and more apt to the current data. It is less frequently used, but see Walther & Nims 2015 Estuaries & Coasts. Nice job. One general thought: this paper is a nice example of how practitioners who analyze a broad suite of elements (AKA the "Kitchen Sink" approach) may likely find that the vast majority are useless for geographic assignment purposes. That's one of my primary takeaways from this current work.	Reviewer 2's previous comments helped us improve our paper. We are very grateful for the constructive comments. We have taken this suggestion on board and will adopt the term "chemoscape" throughout the manuscript, replacing "isoscape". This was always a slightly awkward fit and we agree that while chemoscape is less often used it is clearer for the reader.
Reviewer 3	
Excellent work by the authors to address the comments of all three reviewers. This iteration has been greatly improved and I believe it to be suitable for publication. As I said in my previous review, this is a novel application of a traditional 'isoscape' approach to assess fish connectivity,	The authors are very grateful for the thorough review from reviewer 3. The suggested changes were incredibly useful and have improved the paper greatly.

using otolith element concentrations (instead of the more traditional isotope ratios). Given that the costs associated with analyzing element concentrations in otoliths is typically much lower than the costs associated with measuring otolith isotope ratios, they are often the 'tracer of choice', yet the methods used to analyze the resulting data have not evolved as quickly as those used in isotope movement ecology. As such this paper will have a major influence in this field and provide fisheries and ecosystem managers with new tools and approaches to help protect marine and coastal resources.	
Specific Comments	
1. L73 = affect	1. We think checked all “affect” and “effects”. (Now line 77 after removing track changes.) Checked to use affect as the verb rather than the noun “effect”
2. L74 = I think you should be fully candid here and err more on the side of caution. Physiological effects could augment stock-specific differences (e.g. fast growing stocks vs. slow growing stocks) but they could also confound results (e.g. fast growing individuals within a stock being misclassified to a geographically distant fast-growing stock), particularly if every source hasn’t been extensively sampled. I suggest modifying the next couple of sentences to something like.. “Physiological influences have also been shown to affect element concentrations in the endolymph and otoliths, potentially complicating their use to recover geographic or environmental information. However, so long as all sources have been adequately sampled and samples are cohort matched to avoid potentially confounding effects of interannual variability, the combined effects of physiology and environment may augment among-site	2. We have changed the wording here to incorporate the suggestion. Now line 80 on.

differences in otolith microchemistry and thus increase the power of this tool to reconstruct connectivity patterns. Specifically, elemental concentrations....”	
3. L91 – the wording sounds a bit strange – would this work instead? “...has operated further offshore since the 1990s following declines in gadoid abundance.”	3. Now line 102. We have adopted the suggested wording which does make the sentence clearer.
4. L217 – can you help the reader (e.g. use a non-specialist term here)? I had never heard of a “custom dictionary” in this context.	4. We have changed this to, “...select the optimal predictive set of elements...”
5. L218-219 and 243-247 - Here and in the methods section, it is still unclear to me if you used independent samples to generate the accuracy estimates provided in Lines 243-247. Those values (e.g. “242 km between the predicted catch locations and the true catch locations”) are what the reader will think the accuracy of your adult assignments are. But I believe they will overestimate model prediction power if they were generated using in-sample predictions (fitted values) instead of only out-of-sample predictions. It sounds like you did split the juvenile data and use Monte Carlo cross-validation for model selection (which is great - Line 567-572). But I couldn’t quite work out if you continued to use the split sample when comparing predicted vs known catch locations to assess the final model’s predictive power (i.e. fitted values from the 60% of juvenile samples used to train the model vs. predicted values from the 40% remaining 'independent' samples)? Please clarify clearly here and in the Methods. If you did include fitted values I would caveat the accuracy estimates provided in L243-247 as potentially over optimistic if directly applied to the adult assignments.	5. The accuracy was estimated for each value of precision (0-1 proportion of the chemoscape) for the model during each test run. IE. On run 1 of 500, the data is split, model fitted, and accuracy calculated. Run 2 of 500, the data is split, model fitted, and accuracy calculated and so on... Apologies for the pseudo-code but its maybe easier to describe that way. We have added a fuller description at line 586 to try to make this clearer.

Ideally (as you point out in the discussion – thanks for adding this) – you would have an incredibly extensive juvenile reference library and leave entire sites out from the model fitting procedure to use as independent test samples. You could perform multiple model fitting exercises subsampling individuals and sites to (1) assess whether using a continuous surface approach is totally valid for physiologically sensitive tracers such as otolith element concentrations and/or (2) whether these tracers simply require a more extensive reference libraries than the more traditional isoscapes (where isotope ratios vary predictably with environmental covariates). But this is all food for thought and will hopefully become your follow up study!